# DC-BENCH: Dataset Condensation Benchmark

**Justin Cui[1], Ruochen Wang[1], Si Si[2], Cho-Jui Hsieh[1]**
[1]Department of Computer Science, UCLA,   [2]Google Research
{justincui, ruocwang}@ucla.edu   sisidaisy@google.com   chohsieh@cs.ucla.edu

## Abstract

Dataset Condensation is a newly emerging technique aiming at learning a tiny dataset that captures the rich information encoded in the original dataset. As the size of datasets contemporary machine learning models rely on becomes increasingly large, condensation methods become a prominent direction for accelerating network training and reducing data storage. Despite numerous methods have been proposed in this rapidly growing field, evaluating and comparing different condensation methods is non-trivial and still remains an open issue. The quality of condensed dataset are often shadowed by many critical contributing factors to the end performance, such as data augmentation and model architectures. The lack of a systematic way to evaluate and compare condensation methods not only hinders our understanding of existing techniques, but also discourages practical usage of the synthesized datasets. This work provides the first large-scale standardized benchmark on Dataset Condensation. It consists of a suite of evaluations to comprehensively reflect the generability and effectiveness of condensation methods through the lens of their generated dataset. Leveraging this benchmark, we conduct a large-scale study of current condensation methods, and report many insightful findings that open up new possibilities for future development. The benchmark library, including evaluators, baseline methods, and generated datasets, is open-sourced[1] to facilitate future research and application.

## 1 Introduction

Dataset plays a central role in the performance of machine learning models. With advanced data collection and labeling tools, it becomes easier than ever to construct large scale datasets. The rapidly growing size of contemporary datasets not only posts challenges to data storage and preprocessing, but also makes it increasingly expensive to train machine learning models and design new methods, such as architecture, hyperparameter, and loss function [2, 15, 52, 5]. As a result, data condensation emerges as a promising direction that aims at compressing the original large scale dataset into a small subset of information-rich examples.

In this work, we focus on the newly emerging techniques where the condensed dataset comprises of a set of synthesized samples, learned to matching some statistics to the original dataset or maximizing some utility. In particular, [48] proposed a dataset distillation algorithm to learn the synthesized dataset via bi-level optimization, showing outstanding performance than existing data selection methods under lower compression ratio[2]. After that, many data condensation methods have been proposed to construct synthesized datasets based on various objectives, such as matching gradients [55, 53], embeddings [54], model parameters [4], and kernel ridge regression [32, 33]. It has been reported that all these data-synthesis methods significantly outperform classical data-selection methods.

---

[1]the benchmark is open sourced at `https://github.com/justincui03/dc_benchmark`
[2]compression ratio = compressed dataset size / full dataset size

36th Conference on Neural Information Processing Systems (NeurIPS 2022).

Despite recent efforts in improving dataset condensation, evaluating and comparing different methods is non-trivial and still remains an open issue to date. Previous works [55, 54, 53, 4] mainly evaluate condensation methods by training a randomly initialized model on the condensed dataset and report its test accuracy. During this process, many factors could come into play on the resulting performance, such as data augmentation and architecture. These factors are orthogonal to the condensation algorithms, but might significantly alter their performance. However, there does not exist a unified evaluation protocol among prior works that aligns them. Moreover, the relative performance of condensation methods on real-world downstream applications are also rarely discussed. The lack of a systematic way of evaluation prevents us from establishing fair and thorough comparisons of condensation methods, thereby hinders our understanding of existing algorithms and discourages their practical applications.

This work aims to provide the first benchmark to systematically evaluate data condensation methods through the lens of the condensed dataset. We start by asking the following question: What attributes should a high-quality condensed dataset possess? We identify three main criteria: 1). Models trained on the condensed dataset should achieve good performance across different training protocols, such as different data augmentations and architectures. 2). The condensed dataset should achieve higher performance above naive baseline (random subset of full dataset) across various compression ratios. 3). The condensed dataset should be able to benefit downstream tasks, such as accelerating Neural Architecture Search (NAS). Inspired by these criteria, we propose to measure the strength of condensation algorithms from the following four aspects: 1). *Performance under different augmentation* 2). *Transferability to different architectures* 3). *Performance under different compression ratio* 4). *Performance on NAS task*. These evaluations constitute the core of our benchmark library.

Leveraging the proposed benchmark, we conduct a large-scale comprehensive empirical analysis of state-of-the-art condensation methods. The collection of methods in our library covers four representative dataset condensation methods: Dataset Condensation with Gradient Matching (DC) [55], Differentiable Siamese Augmentation (DSA) [53], Distribution Matching (DM) [55], and Training Trajectory Matching (MTT) [4]. We further include two data-selection methods in our comparisons: random selection and K-Center - a simple algorithm with strong performance. **The experimental results reveal insightful findings on the behavior of condensed dataset, such as:**

- Among all existing methods in our comparison, MTT demonstrates better performance on the proposed evaluation protocols, followed by DSA which often performs the second best, showing promising progressions in the field. However, MTT is not scalable to larger datasets and compression ratio.
- Adding data augmentation to the **evaluation of condensed dataset alone** can significantly boost the performance of all methods.
- Condensation methods are most effective under extremely low compression ratios. As the ratio increases (e.g., CIFAR-10 with 400 images per class), all condensation methods perform similarly to the random selection baseline.
- Despite promising results on training a single model, current condensation methods perform poorly on large-scale real-world tasks, such as Neural Architecture Search and training deep networks beyond the pre-specified architecture. All existing methods encounter similar transferability to the simple K-Center baseline on CIFAR-10 and CIFAR-100.
- Using better data-selection methods (e.g., K-Center) as initialization can drastically improve the convergence speed and performance of condensation methods.

We hope our work could shed lights on deeper understanding of dataset condensation algorithms, designing more comprehensive evaluation, and stimulating future research in advancing the state-of-the-art methods.

## 2 Related Work

### 2.1 Coreset selection methods

Coreset selection method aim to find a representative subset of original dataset [43, 17, 37, 8, 36, 3, 49, 38]. This line of work have enjoyed rich theoretical investigation and a long history of empirical

development, which is worth a separate work. Since our main focus is on dataset synthesis methods, we focus on two commonly used selection-based baselines in dataset condensation literature.

**Random Selection** One naive method of condensation is to randomly pick data from the original dataset, which serves as the baseline for all condensation methods. The performance of random selection is expected to increase steadily as IPC increases, and reach full accuracy when the size of the subset equals the full dataset.

**K-Center** [49, 37, 17] Another commonly used selection based coreset method is K-Center where multiple center points of a class are selected based on a distance function $\mathcal{L}$ so that the distance between data points and their nearest center point is minimized.

## 2.2 Dataset condensation methods

Dataset condensation methods aims to synthesize a small set of data. When it is used for training, competitive performances can be achieved compared to training with the whole dataset. Below we introduce five representative state-of-the-art methods with each using a different technique.

**DC - Dataset Condensation with Gradient Matching [55]** It proposes to infer the synthetic dataset by matching the optimization trajectory of a model trained on synthetic dataset to that on the original dataset. The optimization trajectory is defined as the gradient direction along SGD steps and the loss function to optimize is shown in Equation 1 where $\mathcal{S}$ is the synthetic dataset, T is the number of iterations, $\mathcal{T}$ is the real dataset and $\theta$ are model parameters.

$$\min_{\mathcal{S}} E_{\theta_0 \sim P_{\theta_0}} \Big[ \sum_{t=0}^{T-1} D(\nabla_\theta \mathcal{L}^{\mathcal{S}}(\theta_t), \nabla_\theta \mathcal{L}^{\mathcal{T}}(\theta_t)) \Big] \tag{1}$$

**DSA - Dataset Condensation with Differentiable Siamese Augmentation [53]** DSA proposes to apply Differentiable Siamese Augmentation [56] while learning synthetic image, resulting in more informative synthetic images. Similar to the loss function of DC in Equation 1, DSA applies $\mathcal{A}$ which is a family of image transformations that preserves the semantics of the input as shown in Equation 2.

$$\min_{\mathcal{S}} D(\nabla_\theta \mathcal{L}(\mathcal{A}(\mathcal{S}, \omega^{\mathcal{S}}), \theta_t), \nabla_\theta \mathcal{L}(\mathcal{A}(\mathcal{T}, \omega^{\mathcal{T}}), \theta_t)) \tag{2}$$

**DM - Dataset Condensation with Distribution Matching [54]** Unlike DC and DSA, DM learns condensed dataset by directly matching the output features between real and synthetic samples. The features are acquired from a ConvNet model with randomized weights, which corresponds to data distribution in a randomly projected embedding space. The objective function is shown in Equation 3 where $\psi_v$ is a family of parametric functions to map the input into a lower dimensional space and $\omega \sim \Omega$ is the augmentation parameter.

$$\min_{\mathcal{S}} E_{v \sim P_v, \omega \sim \Omega} \parallel \frac{1}{|\mathcal{T}|} \sum_{i=1}^{|\mathcal{T}|} \psi_v(\mathcal{A}(x_i, \omega)) - \frac{1}{|\mathcal{S}|} \sum_{i=1}^{|\mathcal{S}|} \psi_v(\mathcal{A}(x_i, \omega)) \parallel^2 \tag{3}$$

**MTT - Dataset Distillation by Matching Training Trajectories [4]** is a recently proposed condensation method that builds a condensed dataset to match the parameter trajectory of full data set training. It first updates model parameters using Equation 4 where $D_{syn}$ is the synthetic dataset

$$\hat{\theta}_{t+n+1} = \hat{\theta}_{t+n} - \alpha \nabla \ell(\mathcal{A}(D_{syn}); \hat{\theta}_{t+n}) \tag{4}$$

Then it optimizes the following loss shown in Equation 5 where $\hat{\theta}_{t+N}$ are the student parameters and $\theta_{t+M}^*$ are the future expert parameters.

$$\mathcal{L} = \frac{\parallel \hat{\theta}_{t+N} - \theta_{t+M}^* \parallel_2^2}{\parallel \theta_t^* - \theta_{t+M}^* \parallel_2^2} \tag{5}$$

**KIP - Dataset Meta-Learning from Kernel Ridge-Regression [32, 33]** It performs the condensation process using a new algorithm called Kernel Inducing Point(KIP) through approximating neural networks with kernel ridge-regression(KRR). The objective function is shown in Equation 6 where $K_{UV}$ is the matrix of kernel elements$(K(u, v))_{u \in U, v \in V}$ if U and V are sets. $(X_s, y_s)$ is the support dataset and $(X_t, y_t)$ is the target dataset.

$$L(X_s, y_s) = \frac{1}{2} \parallel y_t - K_{X_t X_s}(K_{X_s X_s} + \lambda I)^{-1} y_s \parallel_2^2 \tag{6}$$

## 2.3 Existing benchmarks

To the best of our knowledge, DC-Bench is the first comprehensive benchmark for dataset synthesis methods. Cross-architecture performance is evaluated in these work [54, 56, 4, 33]. However, the networks used are either too small or too similar. Neural Architecture Search is performed in [55, 56, 54] with different search spaces and methods and it's missing in [4, 33]. With DC-Bench covering all these aspects and beyond with standardized procedures, we believe it will provide useful insights into existing methods and guide future research directions.

# 3 Dataset Condensation Benchmark

In this section, we will layout the details of our benchmark for evaluating different condensation methods. Section 3.1 introduces our proposed evaluation protocols. Section 3.2 explains our implementation details for each method.

In terms of evaluation datasets, we mainly consider three standard image datasets - CIFAR-10, CIFAR-100, and TinyImageNet. These datasets are widely adopted in prior works on dataset condensation. Note that we do not include smaller and simpler datasets such as MNIST [25] or FASHION-MNIST [50], because the performance of different methods are quite similar on these datasets. **CIFAR-10** [22] and **CIFAR-100** [22] both have 50K training images and 10K testing images from 10 and 100 classes. **TinyImageNet** [24, 11] is a subset of the large-scale ImageNet dataset with 200 classes. The training split contains 100K images, and both the validation and test set include 10K images.

## 3.1 Evaluation protocol

In this subsection, we will introduce how we measure the performance of a DC algorithm. More specifically, we will evaluate DC algorithms for four aspects: data augmentation, compression ratio, transferability, and performance on the downstream task of Neural Architecture Search.

### 3.1.1 Performance under different data augmentations

To evaluate the quality of a condensed dataset, a straightforward way is to train a randomly initialized model on it and evaluate the performance (test accuracy) of the model. However, even for the same architecture and condensed dataset, the resulting model can perform very differently under different training protocols, such as the choices of data augmentation and optimizers. Since the condensed datasets are usually very small, we observe that data augmentation methods can significantly impact the test accuracy. Therefore, we investigate the performance of existing condensation methods under four different data augmentation strategies, as listed below.

**ImagenetAug** is a simple manually designed augmentation containing random crop [23], random horizontal flip and color jitters [23]. It is one of the most commonly used strategy for training ImageNet models [28, 19].

**Differentiable Siamese Augmentation (DSA)** is a set of augmentation policies designed for improving the data efficiency of Generative Adversarial Networks (GAN) [16, 35, 42, 56]. The set includes six operations: crop, cutout [12], flip, scale, rotate, and color jitters. At each iteration, one policy is sampled and applied to the input image. DSA is later extended to training synthesized images for dataset condensation [53].

**AutoAugment** [9] is a strong and widely adopted augmentation strategy discovered using AutoML. The searched polices contains a total 16 data augmentations from the popular PIL image library, plus two additional operations: cutout and sample pairing [20]. At each iteration, a pair of augmentations are sampled from the whole policy set and applied onto the search. It achieves state-of-the-art accuracy on CIFAR-10, CIFAR-100, SVHN [31] and ImageNet (without additional data).

**RandAugment** [10] is another popular method of automated policy search, with a different set of discovered policies than AutoAugment. In practice, both RandAugment and AutoAugment are commonly used for training image classification models. Therefore, we also include RandAugment in our benchmark for a more comprehensive evaluation.

*Metrics summary: Similar to real datasets, synthesis datasets should support various augmentations*

*that are tailored to end users' tasks. We evaluate both average and best cases, which could fit different user needs.*

### 3.1.2 Compression ratios

A critical dimension for evaluating dataset condensation methods is their performance under different compression ratios, measured by the number of synthetic Images allocated Per Class (IPC). IPC is typically set by the user of the dataset, according to practical requirements such as storage budget. One question arising naturally is that will dataset synthesis methods continue to compress information effectively under different IPCs? Since the right amount of data for various scenarios might be different (e.g. larger models often require much more data to train), we expect the user to potentially select a wide range of IPCs in practice. As a result, it is crucial to evaluate condensation methods under various compression ratios and identify their effective ranges. However, previous works mainly adopt three IPCs in their evaluation: 1, 10, 50, corresponding to 0.02%, 0.2%, and 1% of the training split in CIFAR-10 [55, 53, 54, 4]; This range is far from comprehensive. In contrast, we evaluate existing condensation methods for up to 1000 IPCs, covering a much wider range of compression ratios. The results under this setting reveal many new insights into the behavior of dataset condensation.

*Metrics summary: As synthesis methods compress info from a large dataset into a small one, we expect a good synthesis method to continue outperforming selection based methods under various compression ratios.*

### 3.1.3 Transferability across architectures

Another dimension of evaluation is on how dataset condensation methods perform across different model architectures. Concretely, a neural network is required to extract statistics from the original and synthetic dataset, and the synthetic dataset is optimized by aligning the extracted information. Therefore, we expect the condensed dataset to perform equally well when it is used to train different architectures. Several previous works [55, 53, 54, 4] provide evaluations under this transfer setting; however the evaluations are mainly with on one dataset or under one IPC. Therefore the resulting conclusion might not generalize to different datasets or IPCs.

To have a deeper understanding for the transferability of condensation methods, we propose a comprehensive protocol that evaluates the performance of condensed datasets under five model architectures, three datasets (CIFAR-10, CIFAR-100, and TinyImageNet), and three IPCs (1, 10, 50). The architectures of choice are as follows:

**MLP:** We first consider a simple MLP architecture. The network includes 3 fully connected layers and the width is set to 128. The number of trainable parameters is around 411K.

**ConvNet [23, 39, 40]:** This is the standard architecture used for both training and evaluating synthetic dataset in previous condensation works. The default network contains three 3x3 convolution layers, each followed by 2x2 average pooling and instance normalization. The hidden embedding size is set to 128. There are around 320K trainable parameters. For TinyImageNet, we increase the number of layers to 4 for improved performance, as suggested in previous work [54, 4].

**ResNet18, ResNet152 [18]:** They are commonly used ResNet architecture with 4/50 residual blocks respectively. Each block contains 2 convolution layers followed by ReLU activation and instance normalization (IN) [44]. There are 11M/60M trainable parameters in ResNet18/ResNet152.

**ViT [14]:** Vision Transformer is a new model architecture that's completely different convolutional networks. It splits an image into fixed-size patches and applies a standard transformer [45] encoder on it. ViT achieves competitive results compared to state-of-the-art convolutional networks with far less computational resource. There are around 10M trainable parameters in our implementation of ViT.

*Metrics summary: Similar to real datasets, a high quality synthetic dataset should be able to be used for training models with various architectures.*

### 3.1.4 Neural Architecture Search

One of the most promising application of data condensation methods is accelerating Neural Architecture Search (NAS) [26, 47, 51, 27, 7, 34, 46, 6, 19]. The goal of NAS is to automatically search for a

top architectures from a vast search space. As a result, the search process typically requiring training and evaluating hundreds or thousands of candidate architectures to obtain their relative performance, which consumes a lot of computation resources [57]. Since the condensed dataset are much smaller than the original dataset, it can potentially be used to accelerate candidate training for NAS algorithms [55]. In the ideal case, the condensed dataset, when deployed to training architectures, can accurately reflect the relative strength of architectures. Concretely, the ranking order (Spearman correlation) of models trained on the condensed dataset should match those trained on the original dataset. To effectively evaluate the condensed dataset, we propose to adopt NAS-Bench-201 [13], a large-scale benchmark database consisting of the ground-truth performance of 15,625 architectures of relatively large size (17 layers). This is different from previous works [55] that only experimented with toy search space made of 720 simple ConvNet architectures.

*Metrics summary: With the primary goal of helping model development by accelerating training, a high quality synthetic dataset should preserve model ranking in Neural Architecture Search task.*

### 3.2 Implementation details

#### 3.2.1 Method of selection

We include 5 state of the art dataset condensation methods: DC, DSA, DM, MTT and KIP and 2 baselines in condensation literature: random selection and K-Center.

**Random Selection** For random selection baseline, we uniformly sample a fixed number of images per class (IPC) from the original dataset.

**K-Center** We use similar approach as [55] where we train a model on the whole dataset to extract features from each data point and use $l_2$ distance to compute class centers. However, we just train the model for 1 epoch using the whole dataset. Our results in Table 1 show that it outperforms all coreset methods used in [55, 54, 56, 4] under almost all settings.

**DC, DSA, DM, MTT** We use the default settings provided by the authors. The only change we made for DC and DSA is that when generating larger IPCs, we update the synthetic dataset per class instead of updating all classes at once as suggested by the author.

**KIP** Since KIP's code is not released, we use the released dataset by the author which are generated under the settings of ZCA preprocessing and no label learning at iteration 1000.

#### 3.2.2 Combining dataset selection with condensation methods

All dataset condensation methods require initializing the synthetic data with either random noise or selected real images. This posts a natural way of combining dataset selection with condensation methods to initialize condensed dataset with the results from data selection. We experiment with three initialization strategies: using 1). *Random selection*, 2). *K-Center*, and 3). *Gaussian noise*. In contrast to prior finding that observes similar results among different initialization methods, we show that (Section 4.6) initialization plays an important role in accelerating the convergence of Dataset Condensation algorithms.

## 4 Empirical studies

### 4.1 Experimental setup

Following previous works, we use ConvNet (Section 3.1.3) to generate the condensed dataset for all our experiments. We follow the default hyperparameters and training configurations of the considered methods, with two exceptions: 1). For IPCs above 50, we manually set and tune the number of iterations for outer and inner optimization for DC and DSA, as they are undefined for large IPCs in the original papers. 2). MTT sometimes applies ZCA whitening as a preprocessing step during both synthetic image training and evaluation, and reported mixed results [4]. Since this is an orthogonal trick that can be applied to any condensation algorithms, we disable it as indicated by the author [4] that it helps convergence and is not crucial to the performance. After the condensed dataset is generated, we train 5 randomly initialized network on it for 1000 epochs using SGD optimizer. **Due to space limit, we only highlight and discuss the cases that lead to most insightful findings in this section. We refer to readers to the full set of results in the appendix.**

Table 1: Test accuracy for random selection, K-Center, DC, DSA, DM, and MTT under different augmentation settings on CIFAR-10, CIFAR-100 and TinyImageNet. The performances without augmentation, best and average performances with augmentation are reported.

| Dataset | IPC | Random | | | K-Center | | | DC | | | DSA | | | DM | | | KIP | | | MTT | | | Whole Dataset |
|---|---|---|---|---|---|---|---|---|---|---|---|---|---|---|---|---|---|---|---|---|---|---|---|
| | | n/a | avg | best | n/a | avg | best | n/a | avg | best | n/a | avg | best | n/a | avg | best | n/a | avg | best | n/a | avg | best | best |
| CIFAR10 | 1 | 15.06 | 14.71 | 15.4 | 23.34 | 22.89 | 25.16 | 28.08 | 26.24 | 29.34 | 27.75 | 26.26 | 27.76 | 26.15 | 24.25 | 26.45 | 35.78 | 29.75 | 40.55 | *39.30* | 31.89 | **44.19** | |
| | 10 | 25.67 | 28.03 | 31.00 | 36.43 | 38.04 | 41.49 | 44.43 | 47.11 | 50.99 | 43.54 | 47.58 | 52.96 | 42.45 | 45.70 | 47.64 | 46.14 | 39.18 | 47.23 | *53.49* | 56.38 | **63.66** | 85.95 |
| | 50 | 44.59 | 47.60 | 50.55 | 48.71 | 53.06 | 56.0 | 53.29 | 54.54 | 56.81 | 54.25 | 56.88 | 60.28 | 56.54 | 60.04 | 61.99 | 53.22 | 52.37 | 56.94 | *62.24* | 65.90 | **70.28** | |
| CIFAR100 | 1 | 4.28 | 4.77 | 5.30 | 8.59 | 9.47 | 10.89 | 12.55 | 12.76 | 13.66 | 13.03 | 12.72 | 13.73 | 10.79 | 8.83 | 11.20 | 6.74 | 8.65 | 12.04 | *16.69* | 13.84 | **22.3** | |
| | 10 | 14.53 | 16.78 | 18.64 | 20.73 | 23.19 | 25.04 | 25.36 | 26.94 | 28.42 | 27.12 | 29.49 | 32.23 | 25.40 | 27.64 | 29.23 | 22.45 | 24.89 | 29.04 | *31.76* | 33.14 | **38.18** | 56.69 |
| | 50 | 29.50 | 32.79 | 34.66 | 33.61 | 36.59 | 38.64 | 29.74 | 27.17 | 30.56 | 38.58 | 40.58 | 43.13 | 37.70 | 40.57 | 42.32 | - | - | - | *43.04* | 42.86 | **46.32** | |
| TinyImageNet | 1 | 1.42 | 1.49 | 1.65 | 2.68 | 2.65 | 3.03 | 5.26 | 4.57 | 5.27 | 5.48 | 5.12 | 5.67 | 3.73 | 3.58 | 3.82 | - | - | - | 5.88 | 6.19 | **8.27** | |
| | 10 | 4.70 | 6.00 | 6.88 | 7.83 | 9.90 | 11.38 | 11.20 | 11.20 | 12.83 | 12.43 | 14.38 | 16.43 | 12.06 | 12.69 | 13.51 | - | - | - | *13.60* | 17.32 | **20.11** | 39.83 |
| | 50 | 13.98 | 16.86 | 18.62 | 16.72 | 20.90 | 22.02 | 11.19 | 10.89 | 12.66 | *21.41* | 22.69 | 25.31 | 20.93 | 21.57 | 22.76 | - | - | - | 20.12 | 25.82 | **28.16** | |

*Numbers* in italic highlights the best accuracy without augmentation. Numbers with underline shows the best average accuracy under different augmentations. **Numbers** in bold represents the highest accuracy with the best augmentation.

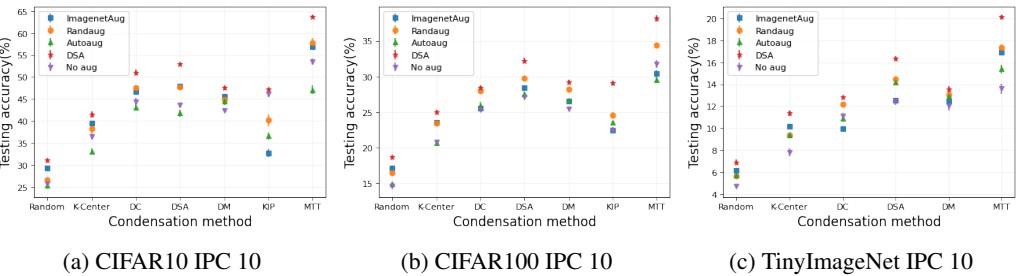

| (a) CIFAR10 IPC 10 | (b) CIFAR100 IPC 10 | (c) TinyImageNet IPC 10 |

Figure 1: Test accuracy for different methods with different augmentations.

## 4.2 Data augmentation

We start by evaluating dataset condensation methods using our benchmark with five augmentation strategies: DSA, AutoAugment, RandAugment, ImagenetAug, and no augmentation. Note that our empirical study is conducted from the end user perspective: i.e. we treat the generation of condensed dataset as blackbox, and apply different augmentations during the evaluation of the condensed datasets The results are visualized in Figure 1. We observe that **applying the right data augmentation during evaluation significantly improves the performance** of model trained on the condensed dataset. For example, on CIFAR-10 and IPC=10, the test accuracy of DC, DSA, DM and MTT increases by 6.56%, 9.4%, 5.2% and 10.2% respectively.

One interesting comparison is between DC and DSA. Their only difference is that DSA applies augmentation during both synthetic dataset training and evaluation, whereas DC does not use any augmentation [55, 53]. In our experiment, we found that **the performance of DC is largely underestimated, due to misalignment in the evaluation protocol**. After applying the best augmentation, DC's performance increases by up to 6%. On the other hand, when data augmentation is disabled during evaluation, the performance of condensed dataset trained by DSA drops to a similar level of DC in many cases.

Although in some cases users may stick with the best augmentation associated with the condensed dataset, there exists cases that user wish to use different augmentations depending on the task at hand. Therefore, we propose to evaluate each method under: 1). no augmentation 2). best augmentation and 3). the average test accuracy under all augmentations. The numerical results are summarized in Table 1. We observe that: **1). DSA augmentation overall is the best strategies in our experiments. 2). Some methods exhibit much higher variance across different augmentation.** For example, although DSA and MTT achieves top 2 performance, they have a much wider accuracy range than DM. This indicates that DSA and MTT might potentially overfit to the augmentation used during synthetic dataset learning.

Further, to establish fair comparison, we also reevaluate Data-Selection methods with augmentation enabled. This has been overlooked in many previous works where the baseline data-selection methods they compared against are all evaluated and reported without augmentation. As shown in Table 1, similar to Data-Synthesis methods, **Data-Selection method also benefits significantly from augmentation**. For instance, both random selection and K-Center achieves over 5% absolute gain in test accuracy on all dataset under IPC 50 (except for TinyImageNet which is close: 4.64%). Most noticeably, under 50 IPCs, **K-Center even outperforms DC on CIFAR-100 and TinyImageNet,**

**and on-par with DC on CIFAR-10.** The impressive performance of K-Center showcases that the potential of selection-based methods is drastically underestimated in previous works.

**Key Takeaways 1:** Augmentation applied during evaluation alone can drastically increase model performances for both selection and condensation methods. This also causes several previous methods to be largely underestimated.

**Key Takeaways 2:** Some methods are particularly sensitive to data augmentation(e.g.KIP, MTT). Regardless of the augmentations applied during data generation, downstream tasks still have to try out different types of augmentations in order to find the right augmentation that achieves the best results.

### 4.3 Different compression ratios

As previous discussed, prior works mainly evaluate condensed dataset of size up to 50 IPCs (1% compression ratio). However, this is a rather extreme case, and in most practical applications users may be willing to increase the compression ratio to get a more informative subset. To analyze the performance of condensation algorithms under larger compression ratios, we rerun the selected methods with IPC up to 1,000. For fair comparison, DSA augmentation is enabled during evaluation for all base methods, and the synthetic dataset is initialized from random real images.

Ideally, we expect the performance trajectory of condensation methods to approach the oracle accuracy (brown horizontal line) on the full dataset much faster than the random selection. However, this is not the case in practice. As shown in Figure 2, the performance gain of dataset condensation methods over selection based methods is only obvious for IPCs less than 200. **As IPC increases further**, the margin shrinks considerably and **all methods perform similarly to the random selection baseline**. Considering the fact that the synthetic datasets are initialized from random selection in the first place, it reveals that all current condensation methods fail to effectively explore the increased capacity brought by the extra IPCs. Moreover, although DM [54] outperforms DSA [53] with IPC 50, it's not able to consistently outperform DSA [53] with larger

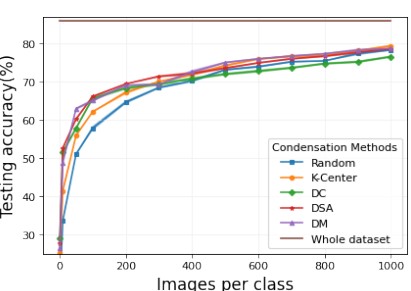

Figure 2: Performance comparison with different compression ratios on CIFAR-10 using Random selection, K-Center, DC, DSA and DM.

IPCs which contradicts with the claim made in [54]. We **do not include MTT in the plot as it does not scale well to IPCs beyond 50 on CIFAR-10, in terms of both memory and run-time.** The scalability becomes worse on larger dataset with more classes and higher resolutions. For ease of reference, we also report numerical results in Appendix Table 6.

**Key Takeaways 1:** Condensation methods perform better than selection methods under small IPCs.
**Key Takeaways 2:** When IPC goes above 200, synthesis based methods' performance degrades to that of random selection baselines

### 4.4 Transferability

We evaluate the transferability of different condensation methods on the 3 datasets with IPC 1, 10 and 50 using the 5 architectures described previously. As showing in Figure 3, the performance of all synthetic methods drops when transferring to other architectures. Moreover, **the relative ranking of different condensed dataset might not necessarily be preserved when transferred to different architectures**. For instance, on CIFAR-100, while DC outperforms K-Center on ConvNet, it falls behind on ResNet architecture. Another example is on CIFAR-10, where we see that the dataset learned by the top-notch method, MTT, transfers poorly to MLP networks. In addition, we observe that **K-Center achieves the better transferability than several condensation methods** (Table 2).

One particular observation we want to point out is that **none of the condensation methods we tested perform well when transferred to large models like ResNet152**. We conjecture that it is because the larger ResNet152 might require more data to optimize than models of much smaller size, such as ConvNet. This is evidenced by the fact that ResNet152 performs worse than any other architecture on random selection with small IPCs. To further verify it, we trained ResNet152

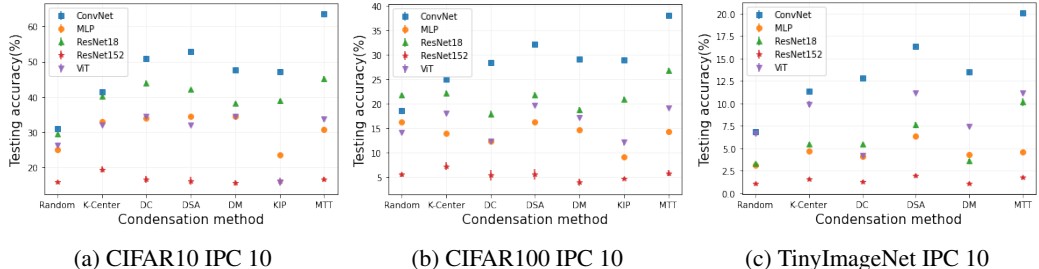

| (a) CIFAR10 IPC 10 | (b) CIFAR100 IPC 10 | (c) TinyImageNet IPC 10 |

Figure 3: Synthetic dataset performance evaluated using different networks.

with 300 randomly selected images per class and obtain an accuracy of 45.59%; When the IPC reaches 1,000, ResNet152 achieves 70.29% accuracy, gradually gaining back its full potential.

This result serves as an extra piece of evidence that a good condensed method should operate robustly under different compression ratios. Please refer to appendix for the numerical results.

**Key Takeaways 1:** Dataset synthesis methods' performance drops on other architectures.

**Key Takeaways 2:** None of the condensation methods transfer well to large models such as ResNet152.

**Key Takeaways 3:** The relative ranking of different methods may not be preserved when transferring to different architectures.

Table 2: Testing accuracy of different methods on ConvNet versus transferred to other architectures. The "Transfer" column records the average results on MLP, ResNet18, ResNet152 and ViT. All methods are evaluated with 10 IPCs. Results under other IPCs can be found in the Appendix.

|  | CIFAR-10 | | CIFAR-100 | | TinyImageNet | |
|---|---|---|---|---|---|---|
|  | ConvNet | Transfer | ConvNet | Transfer | ConvNet | Transfer |
| Random | 31.00 | 24.16 | 18.64 | 11.31 | 6.88 | 3.53 |
| K-Center | 41.19 | 31.01 | 25.04 | 15.31 | 11.38 | 5.42 |
| DC | 50.99 | **32.22** | 28.42 | 11.95 | 12.83 | 3.74 |
| DSA | 52.96 | 31.15 | 32.23 | 15.77 | 16.34 | 6.75 |
| DM | 47.64 | 30.66 | 29.23 | 13.59 | 13.51 | 4.08 |
| KIP | 47.23 | 23.54 | 29.04 | 11.76 | - | - |
| MTT | **63.66** | 31.55 | **38.18** | **16.48** | **20.11** | **6.91** |

## 4.5 Neural Architecture Search (NAS)

One promising application of DC is on Neural Architecture Search as shown in [55]. To evaluate this task, we randomly sample 100 networks from NAS-Bench-201 [13], which contains ground-truth performance of 15,625 networks. All models are trained on CIFAR-10 for 50 epochs under 5 random seeds, and ranked according to their average accuracy on a held-out validation set of 10k images.

We reduce the number of repeated blocks from 15 to 3 during the search phase, as we found that original networks perform poorly when trained on condensed dataset due to their size. This is a common practice in NAS [28]. We measure the performance on task NAS with two metrics: 1). Correlation between the ranking of models trained on condensed dataset and original dataset 2). The ground-truth performance of the best architecture trained on the condensed

Table 3: Spearman's rank correlation using NAS-Bench-201. The state-of-the-art performance on the test set is **94.36%**. The rank correlation of original dataset is lower than 1.0 because we use a small architecture and perform ranking based on validation set.

|  | Random | K-Center | DC | DSA | DM | KIP | MTT | Original Dataset |
|---|---|---|---|---|---|---|---|---|
| Correlation | -0.06 | 0.11 | -0.19 | -0.37 | -0.37 | -0.50 | -0.09 | 0.7487 |
| Top 1 (%) | 91.9 | 91.78 | 86.44 | 73.54 | 92.16 | 92.91 | 73.54 | 93.5 |

dataset (Top 1). We argue that ranking correlation is more important than Top 1 accuracy, as it measures how well the condensed dataset can reflect the relative strength of various architectures [1].

Although prior work on utilizing condensed dataset for NAS reports promising results (0.79 Spearman correlation using DC [55]), they mainly consider toy search spaces made of 720 ConvNet architectures. With larger scale NAS benchmark with modern architectures, we have a different observation. As shown in Table 3, we found that **there is little or even negative correlation between the performance on condensed and full dataset**. All methods produce negative correlation except for K-Center. This shows that condensed dataset fails to preserve the true strength of the underlying model. Moreover, the best architecture discovered from DC and DSA's condensed dataset performs poorly on the full dataset. Our result indicates that, despite the performance gain brought by recent condensation methods on training a single specific model, it remains challenging to truly utilize the condensed dataset to guide model designs.

**Key Takeaways 1:** Although some previous works show promising results on toy networks, none of them are suitable for standardized neural architecture search tasks.

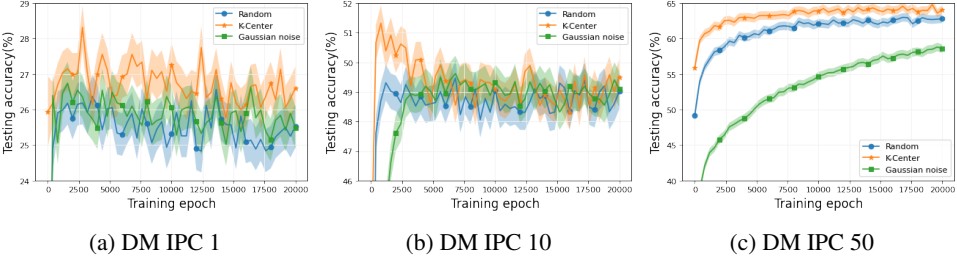

| (a) DM IPC 1 | (b) DM IPC 10 | (c) DM IPC 50 |

Figure 4: Test accuracy comparison among initializing synthetic images with Random selection, K-Center and Gaussian noise on CIFAR-10 using DM.

**Key Takeaways 2:** As accelerating model training is one of the major use cases of condensed dataset, we encourage the community to incorporate this standardized NAS task popularized by modern architectures.

### 4.6 Combining Data-Selection methods with Data-Synthesis methods

Initialization of synthetic dataset is a relatively unexplored territory. Prior Data-Synthesis methods typically initialize the dataset from either Gaussian noise or randomly selected images. Since advanced selection methods such as K-Center outperforms random selection by a large margin, a natural question is whether condensation methods would benefit from images selected from K-Center. This also serves as a direct way of combining data selection methods with data synthesis methods. We test our hypothesis by using images from K-Center to initialize condensation methods. As shown in Figure 4, K-Center initialization not only converges faster, but also achieves improved end performance in some cases. On average, we observe that K-Center only requires about 30% of the computation budget to reach the same level of performance as random selection; The end performance is also 1.3% higher as well. The saving is desirable, especially considering the fact that the condensed dataset usually takes a long time to train (e.g. 15h for DSA under 50 IPCs on CIFAR-10). Due to space limit, we only show the curves of DM in the main text; the plots of other methods can be found in the appendix.

**Key Takeaways:** Synthetic data initialization plays an crucial role in the convergence and final performance of condensation methods.

## 5 DC-BENCH library

We design and implement an evaluation library that incorporates all the aforementioned protocols. To facilitate the research on dataset condensation, we set up a leaderboard (`https://dc-bench.github.io/`) to record the performance of existing condensation methods and new submissions. The entire benchmark, including the evaluation library, condensed datasets, and scripts to reproduce the results in this paper, can be found at (`https://github.com/justincui03/dc_benchmark`). **Both the leaderboard and the benchmark will be updated regularly to reflect the most recent progress in dataset condensation methods**.

## 6 Outlook

**Conclusion** This paper introduces the first large-scale benchmark on dataset condensation methods. Leveraging the proposed benchmark, we conduct the first comprehensive empirical analysis of existing condensation algorithms. Our study reveals several scenarios where current method can be improved, leading to the following potential research directions: 1). (automated) designing better augmentation that suits the synthetic data, 2). improving the transferability of condensed dataset to other architectures 3). condensation methods for NAS 4). developing condensed methods that perform well for a wide range of compression ratios. and 5) effective ways to combine Data-Selection with Data-Synthesis methods. We hope the proposed benchmark could guide users to choose and evaluate various DC methods and facilitate future developments of advanced data condensation methods. **Limitation and outlook** While the current iteration of our benchmark is comprehensive at the moment, it could become limited in scope as the field advances. In the future, we plan to expand the scope of our DC-Bench to include more architectures, datasets, and downstream tasks. As current methods primarily focus on image classification, we will also incorporate tasks from other modality, such as text, graph, and audio.

## Acknowledgments and Disclosure of Funding

This work is supported in part by NSF under IIS-2008173 and IIS-2048280. CJH is also supported by research awards from Google and the Okawa Foundation.

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
