# A Appendix

## A.1 Hyperparameters

For all synthesis based methods, we use the default parameters given by the authors in their original paper. E.g. For DC and DSA, we run it for 1000 iterations. The only change we make in order to have them scale up to IPCs larger than 50 is setting inner loop and outer loop to both 10 which are not given by the original authors. For DM, we run the condensation method for 20,000 iterations which is the same as the author. For MTT, we use the condensed dataset provided by the author whose exact settings can be found in [4]. For K-Center, we use the Kmeans implementation inside scikit-learn which is a publicly available library with maximum 300 iterations. In order to generate the embedding features for the image, we create a random ConvNet model and train it with 1 epoch using the whole dataset. For CIFAR10/100, we use a 3 layer ConvNet, for TinyImageNet, we use a 4 layer ConvNet as suggested by [54]. For the augmentation used to generate the synthesis dataset, we keep it the same as the original author, e.g. no augmentation for DC, DSA augmentation for DSA, DM and MTT, ZCA preprocessing for KIP.

## A.2 Data augmentation

In Table 1, we report the average and best performance of different methods under different augmentation settings. Here we show the complete numerical results in Table 4

## A.3 Transferability

Besides Figure 3 shown previously which includes the case when IPC equals 10, here we show Figure 5 that contains IPC 1 and 50 for CIFAR-10, CIFAR-100, TinyImageNet. We also include all the numerical results in Table 5 for reader's references.

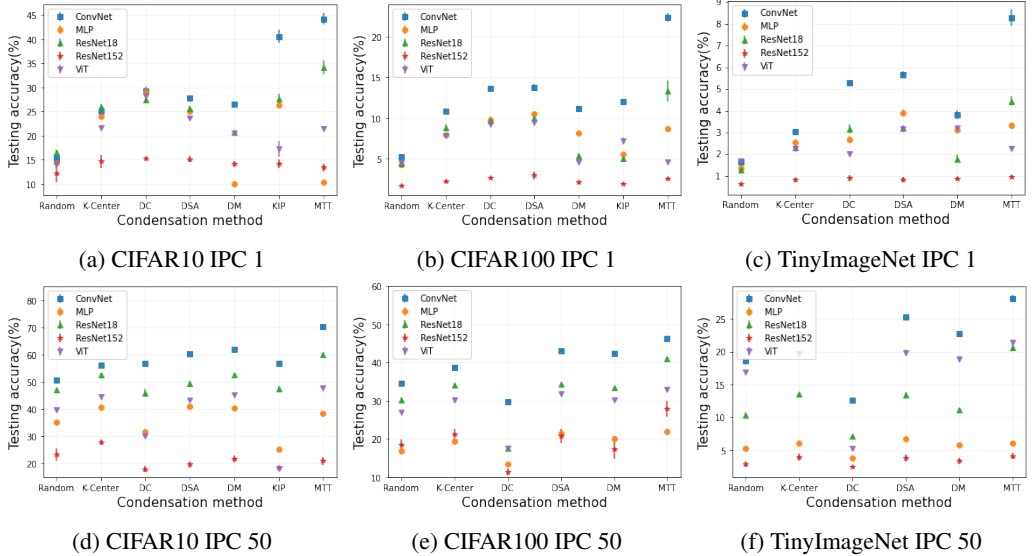

(a) CIFAR10 IPC 1     (b) CIFAR100 IPC 1     (c) TinyImageNet IPC 1

(d) CIFAR10 IPC 50     (e) CIFAR100 IPC 50     (f) TinyImageNet IPC 50

Figure 5: Condensation method transferability for IPC 1 and 50

## A.4 Combining selection based methods with synthesis based methods

As shown in Figure 4 for DM, if we combine synthesis based method with better selection method, it will not only converge faster but also achieve better performance. Similar results from DC and DSA can be seen from Figure 6 on dataset CIFAR-10. **We are not able to get the performance of MTT under the computation resource limit(Out Of Memory) we set.**

**Algorithm 1** K-Center

Randomly initialize a ConvNet model **M** and train for 1 single epoch
Compute the embedding features for each image using model **M**
**for** i in 1 .. **N do**
    Select all the images belonging to class i and assign them to $S_i$
    Randomly select **P** images from $S_i$ and assign to $C_i$ as the centers for KMmeans.
    **while** $j \leq$ **K do**
        update $C_i$ based on the $L_2$ distance in the embedding space.
    **end while**
    Based on the center $C_i$, get the nearest N images using $L_2$ distance in the embedding space
**end for**
**Return** $\{C_i \mid i = 1..N\}$

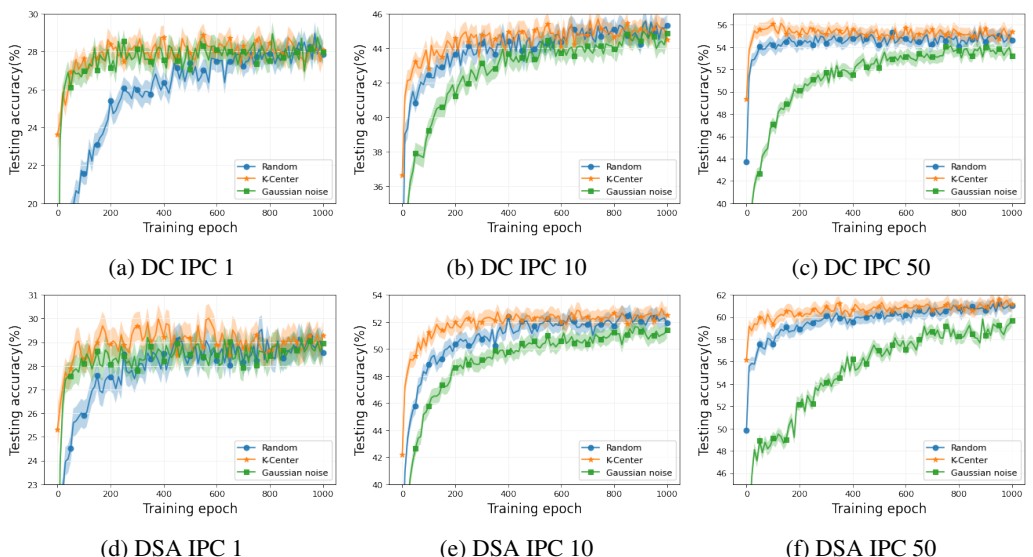

Figure 6: Test accuracy comparison between initializing synthetic images with Random selection, K-Center and Gaussian noise on CIFAR-10

## A.5  Different compression ratios

Previously, we show the condensation method performances in Figure 2 with IPC up to 1000, here we show the numeric results in Table 6 for easier references.

## A.6  Impact of different optimizers

The choice of optimizer (e.g., SGD versus Adam [21]) usually has minor impact to the performance according to our experiments. In Table 7, we show the results of using SGD optimizer versus Adam optimizer with all IPCs(1, 10, 50) on CIFAR-10. All performances are evaluated with DSA augmentation. We get similar performances with the 2 different optimizers.

## A.7  Computation resources

In order to perform a fair comparison, we run all the experiments on the same virtual machine with 1 NVIDIA A100 GPU with 40GB GPU memory on Google Cloud. If some method is not able to run on the machine within GPU memory limit, we report it as out-of-memory(OOM).

## A.8  Training time

Here we show the training time of each condensation methods in Table 8. For K-Center, we iterate 300 times to find the cluster center. The run time is computed by dividing the total time by 300. For

synthesis based methods, we run each method 100 epochs and report the mean and standard deviation of the data.

## A.9 K-Center

Prior condensation works have examined a variety of selection-based methods as the baseline, such as Herding, K-Center, and Forgetting [41]. But surprisingly, these methods often fail to outperform even the random selection baseline. For instance, K-Center, which pick data based on the centers of KMean [29] algorithm, reports 16.4% (absolute difference) lower test accuracy on CIFAR-10 and 50 IPCs than random selection in the original DC paper [55]. After careful investigation, we find that the reason is probably that the model used to extract features is trained for too many epochs, causing the features to be too close for the data in the same class Our revised implementation leads to a substantial performance gain over both random selection and prior selection methods (Table 1,9). In addition, K-Center can even match the performance of some condensation methods in some cases (Table 1) while being much faster to run. We show the flow of it in Algorithm 1

Table 9: Kmeans test accuracy comparison using raw image feature and embedding feature generated by ConvNet on CIFAR10/100 and TinyImageNet. For CIFAR10/100, a 3-layer ConvNet is used. For TinyImageNet, a 4-layer ConveNet is used.

| Dataset | IPC | Image Feature | |
| | | raw input | embedding |
|---|---|---|---|
| CIFAR-10 | 1 | 21.86 | **25.16** |
| | 10 | 32.21 | **41.49** |
| | 50 | 47.00 | **56.00** |
| CIFAR-100 | 1 | 6.74 | **10.89** |
| | 10 | 20.08 | **25.04** |
| | 50 | 35.15 | **38.64** |
| TinyImageNet | 1 | 1.98 | **3.03** |
| | 10 | 6.88 | **11.38** |
| | 50 | 17.85 | **22.02** |

## A.10 Distribution bias of synthetic dataset

Since DC methods synthesize a small dataset, one natural question is whether it introduces any bias into the data distribution. Therefore, we plot the data distribution of the real dataset and the synthetic dataset in Figure 7 for CIFAR-10 with IPC 50. The features are extracted with a ResNet18 model fully trained on the real dataset. ZCA is applied when extracting features for KIP. The figure is visualized using T-SNE [30] for the first class. As we can see that the synthetic images learned by DC and DSA are more on the edge of the distribution. The data learned by DM, KIP and MTT are more towards the center.



| (a) DC | (b) DSA | (c) DM | (d) KIP | (e) MTT |

Figure 7: Synthetic dataset distribution with IPC 50

## A.11 Example real images and synthetic images

Here we show the selected real images generated by Random selection, K-Center and synthetic images generated by DC, DSA, DM and MTT in Figure 8 for reader's references.

## A.12 Source code and leaderboard

We open source our benchmark and evaluation library at https://github.com/justincui03/dc_benchmark as a contribution to the research community. At the same time, we build a leaderboard https://justincui03.github.io/dcbench/ to track the most up-to-date progress in the field. All these will be maintained regularly and we will keep updating the benchmark to reflect the newest changes.

### A.13 Assets license

**Code License**:Our codebase is open sourced under the MIT license.
**Dataset License**: The datasets(CIFAR10/100, TinyImageNet) used in this paper are not part of our assets, if readers are going to use the datasets, please follow the instructions below

- **CIFAR10/100** Please refer to **requirements for usage** for these 2 datasets.

- **TinyImageNet** Please refer to **term of access** for using TinyImageNet.

### Ethics Statements

The condensed dataset used in this paper are all generated from the following standard non-private dataset: CIFAR-10, CIFAR-100, and TinyImageNet. The library itself does not include any sensitive information or components. Therefore, we are not aware of any ethical concern of the benchmark. However, the end users should be aware of potential data leakage through condensed dataset, when they try to apply any condensation methods included in our benchmark to their tasks at hand.

**Appendix resumes on the next page.**

Table 4: Complete test accuracy with variance for Random selection, K-Center, DC, DSA, DM, MTT under different augmentation settings on CIFAR-10, CIFAR-100 and TinyImageNet.

| Dataset | Method | IPC | Augmentation | | | | |
| --- | --- | --- | --- | --- | --- | --- | --- |
| | | | n/a | imagenet_aug | randaug | autoaug | DSA |
| CIFAR-10 | Random | 1 | 15.06 ± 0.60 | 15.07 ± 0.27 | 14.58 ± 0.54 | 13.78 ± 0.69 | 15.4 ± 0.28 |
| | | 10 | 25.67 ± 0.36 | 29.22 ± 0.33 | 26.59 ± 0.58 | 25.32 ± 0.67 | 31.00 ± 0.48 |
| | | 50 | 44.59 ± 0.32 | 50.00 ± 0.48 | 47.36 ± 0.52 | 42.47 ± 0.57 | 50.55 ± 0.32 |
| | K-Center | 1 | 23.34 ± 0.90 | 25.92 ± 1.12 | 21.99 ± 0.53 | 18.5 ± 0.62 | 25.16 ± 0.45 |
| | | 10 | 36.43 ± 0.63 | 39.46 ± 0.59 | 38.18 ± 1.52 | 33.01 ± 0.69 | 41.49 ± 0.73 |
| | | 50 | 48.71 ± 0.33 | 55.53 ± 0.51 | 53.09 ± 0.41 | 47.62 ± 0.53 | 56.00 ± 0.29 |
| | DC | 1 | 28.08 ± 0.80 | 25.70 ± 0.82 | 27.39 ± 0.81 | 22.51 ± 0.91 | 29.34 ± 0.37 |
| | | 10 | 44.43 ± 0.85 | 46.67 ± 0.43 | 47.65 ± 0.36 | 43.13 ± 0.80 | 50.99 ± 0.62 |
| | | 50 | 53.29 ± 0.92 | 55.04 ± 0.20 | 54.82 ± 0.31 | 51.49 ± 0.42 | 56.81 ± 0.44 |
| | DSA | 1 | 27.75 ± 0.54 | 25.81 ± 0.66 | 26.71 ± 0.57 | 24.75 ± 0.98 | 27.76 ± 0.47 |
| | | 10 | 43.54 ± 0.37 | 47.84 ± 0.41 | 47.78 ± 0.57 | 41.75 ± 0.81 | 52.96 ± 0.41 |
| | | 50 | 54.25 ± 0.57 | 59.11 ± 0.57 | 56.15 ± 0.71 | 51.98 ± 0.51 | 60.28 ± 0.37 |
| | DM | 1 | 26.15 ± 0.80 | 24.54 ± 0.67 | 24.96 ± 0.89 | 21.06 ± 2.03 | 26.45 ± 0.39 |
| | | 10 | 42.45 ± 0.43 | 45.67 ± 0.55 | 44.95 ± 0.53 | 44.55 ± 0.97 | 47.64 ± 0.55 |
| | | 50 | 56.54 ± 0.41 | 60.42 ± 0.42 | 60.17 ± 0.43 | 57.56 ± 0.42 | 61.99 ± 0.33 |
| | KIP | 1 | 24.30 ± 1.89 | 28.34 ± 2.16 | 25.79 ± 1.23 | 40.55 ± 1.34 | 35.78 ± 1.03 |
| | | 10 | 32.75 ± 0.91 | 40.13 ± 1.29 | 36.59 ± 0.83 | 47.23 ± 0.40 | 46.14 ± 0.68 |
| | | 50 | 50.73 ± 0.55 | 51.55 ± 0.67 | 50.26 ± 0.46 | 56.94 ± 0.38 | 53.22 ± 0.71 |
| | MTT | 1 | 39.30 ± 1.14 | 28.23 ± 2.09 | 31.57 ± 1.41 | 23.57 ± 1.45 | 44.19 ± 1.18 |
| | | 10 | 53.49 ± 0.74 | 56.86 ± 0.57 | 57.87 ± 0.93 | 47.11 ± 0.98 | 63.66 ± 0.38 |
| | | 50 | 62.24 ± 0.52 | 65.67 ± 0.75 | 66.38 ± 0.44 | 61.25 ± 0.76 | 70.28 ± 0.61 |
| CIFAR-100 | Random | 1 | 4.28 ± 0.20 | 4.67 ± 0.15 | 4.60 ± 0.17 | 4.49 ± 0.13 | 5.30 ± 0.23 |
| | | 10 | 14.53 ± 0.27 | 17.1 ± 0.36 | 16.44 ± 0.30 | 14.93 ± 0.24 | 18.64 ± 0.25 |
| | | 50 | 29.50 ± 0.26 | 34.24 ± 0.23 | 31.94 ± 0.26 | 30.31 ± 0.31 | 34.66 ± 0.41 |
| | K-Center | 1 | 8.59 ± 0.27 | 9.25 ± 0.19 | 9.14 ± 0.23 | 8.60 ± 0.32 | 10.89 ± 0.17 |
| | | 10 | 20.73 ± 0.22 | 23.56 ± 0.19 | 23.48 ± 0.19 | 20.67 ± 0.23 | 25.04 ± 0.30 |
| | | 50 | 33.61 ± 0.41 | 37.73 ± 0.34 | 36.19 ± 0.32 | 33.79 ± 0.37 | 38.64 ± 0.43 |
| | DC | 1 | 12.55 ± 0.37 | 11.46 ± 0.28 | 12.98 ± 0.42 | 12.93 ± 0.24 | 13.66 ± 0.29 |
| | | 10 | 25.36 ± 0.28 | 25.51 ± 0.36 | 27.96 ± 0.30 | 25.87 ± 0.53 | 28.42 ± 0.29 |
| | | 50 | 29.74 ± 0.34 | 24.72 ± 0.34 | 27.54 ± 0.42 | 25.87 ± 0.18 | 30.56 ± 0.56 |
| | DSA | 1 | 13.03 ± 0.14 | 11.50 ± 0.15 | 12.61 ± 0.33 | 13.03 ± 0.33 | 13.73 ± 0.45 |
| | | 10 | 27.12 ± 0.27 | 28.38 ± 0.32 | 29.78 ± 0.29 | 27.58 ± 0.25 | 32.23 ± 0.35 |
| | | 50 | 38.58 ± 0.28 | 40.26 ± 0.34 | 40.81 ± 0.27 | 38.11 ± 0.54 | 43.13 ± 0.33 |
| | DM | 1 | 10.79 ± 0.31 | 7.52 ± 0.22 | 8.75 ± 0.30 | 7.85 ± 0.44 | 11.20 ± 0.27 |
| | | 10 | 25.40 ± 0.22 | 26.5 ± 0.13 | 28.15 ± 0.32 | 26.66 ± 0.25 | 29.23 ± 0.26 |
| | | 50 | 37.7 ± 0.26 | 40.45 ± 0.32 | 40.69 ± 0.35 | 38.81 ± 0.31 | 42.32 ± 0.37 |
| | KIP | 1 | 8.16 ± 0.29 | 7.02 ± 0.28 | 7.36 ± 0.28 | 12.04 ± 0.15 | 6.74 ± 0.33 |
| | | 10 | 22.45 ± 0.44 | 24.52 ± 0.42 | 23.53 ± 0.20 | 29.04 ± 0.34 | 22.45 ± 0.28 |
| | MTT | 1 | 16.69 ± 0.64 | 11.55 ± 0.33 | 10.87 ± 0.34 | 10.65 ± 0.35 | 22.3 ± 0.55 |
| | | 10 | 31.76 ± 0.54 | 30.46 ± 0.37 | 34.37 ± 0.36 | 29.53 ± 0.41 | 38.18 ± 0.42 |
| | | 50 | 43.04 ± 0.48 | 41.55 ± 0.26 | 44.73 ± 0.33 | 38.84 ± 0.26 | 46.32 ± 0.26 |
| TinyImageNet | Random | 1 | 1.42 ± 0.08 | 1.45 ± 0.05 | 1.50 ± 0.08 | 1.37 ± 0.08 | 1.65 ± 0.11 |
| | | 10 | 4.70 ± 0.18 | 6.15 ± 0.11 | 5.66 ± 0.16 | 5.27 ± 0.19 | 6.88 ± 0.25 |
| | | 50 | 13.98 ± 0.28 | 17.39 ± 0.21 | 16.44 ± 0.25 | 15.00 ± 0.32 | 18.62 ± 0.22 |
| | K-Center | 1 | 2.68 ± 0.20 | 2.68 ± 0.18 | 2.53 ± 0.11 | 2.34 ± 0.15 | 3.03 ± 0.12 |
| | | 10 | 7.83 ± 0.35 | 10.17 ± 0.20 | 9.40 ± 0.22 | 8.63 ± 0.18 | 11.38 ± 0.26 |
| | | 50 | 16.72 ± 0.41 | 20.47 ± 0.23 | 19.79 ± 0.46 | 17.75 ± 0.18 | 22.02 ± 0.40 |
| | DC | 1 | 5.26 ± 0.19 | 4.02 ± 0.13 | 4.80 ± 0.09 | 4.18 ± 0.21 | 5.27 ± 0.10 |
| | | 10 | 11.12 ± 0.28 | 9.95 ± 0.24 | 12.16 ± 0.20 | 9.86 ± 0.22 | 12.83 ± 0.14 |
| | | 50 | 11.19 ± 0.27 | 9.26 ± 0.37 | 12.303 ± 0.19 | 9.33 ± 0.13 | 12.66 ± 0.36 |
| | DSA | 1 | 5.48 ± 0.14 | 4.19 ± 0.10 | 5.23 ± 0.32 | 5.38 ± 0.11 | 5.67 ± 0.14 |
| | | 10 | 12.43 ± 0.29 | 12.53 ± 0.24 | 14.46 ± 0.23 | 14.18 ± 0.14 | 16.34 ± 0.21 |
| | | 50 | 21.41 ± 0.25 | 22.47 ± 0.37 | 22.98 ± 0.35 | 20.00 ± 0.29 | 25.31 ± 0.22 |
| | DM | 1 | 3.73 ± 0.22 | 3.15 ± 0.19 | 3.65 ± 0.21 | 3.70 ± 0.16 | 3.82 ± 0.21 |
| | | 10 | 12.06 ± 0.43 | 12.43 ± 0.31 | 13.04 ± 0.18 | 11.76 ± 0.17 | 13.51 ± 0.31 |
| | | 50 | 20.93 ± 0.32 | 22.19 ± 0.34 | 22.03 ± 0.22 | 19.31 ± 0.33 | 22.76 ± 0.28 |
| | MTT | 1 | 5.88 ± 0.41 | 5.13 ± 0.30 | 5.92 ± 0.16 | 5.44 ± 0.23 | 8.27 ± 0.36 |
| | | 10 | 13.6 ± 0.47 | 16.89 ± 0.15 | 17.31 ± 0.31 | 14.97 ± 0.38 | 20.11 ± 0.16 |
| | | 50 | 20.12 ± 0.30 | 25.33 ± 0.31 | 26.49 ± 0.32 | 23.30 ± 0.18 | 28.16 ± 0.45 |

Whole training set performances are: 85.95 ± 0.09 on CIFAR-10, 56.69 ± 0.18 on CIFAR-100 and 39.83 ± 0.41 on TinyImageNet with DSA augmentation.

Table 5: Transferability of different methods using 5 different networks with IPC 1, 10 and 50. All results are evaluated with DSA augmentation.

| Dataset | Method | IPC | ConvNet | MLP | Network ResNet18 | ResNet152 | ViT |
|---------|--------|-----|---------|-----|------------------|-----------|-----|
| CIFAR-10 | Random | 1 | 15.40 ± 0.28 | 14.37 ± 0.38 | 16.56 ± 0.46 | 12.15 ± 1.80 | 14.19 ±0.99 |
| | | 10 | 31.00 ± 0.48 | 25.08 ± 0.27 | 29.52 ± 0.87 | 15.84 ± 0.91 | 26.21 ± 0.49 |
| | | 50 | 50.55 ± 0.32 | 35.21 ± 0.44 | 47.26 ± 0.27 | 23.36 ± 2.31 | 39.73 ± 0.52 |
| | K-Center | 1 | 25.16 ± 0.45 | 24.01 ± 0.32 | 25.99 ± 0.57 | 14.64 ± 1.30 | 21.54 ± 0.55 |
| | | 10 | 41.49 ± 0.73 | 32.92 ± 0.38 | 40.08 ± 0.88 | 19.35 ± 0.71 | 31.95 ± 0.57 |
| | | 50 | 56.00 ± 0.29 | 40.61 ± 0.34 | 52.69 ± 0.70 | 27.84 ± 1.07 | 44.65 ± 0.39 |
| | DC | 1 | 29.34 ± 0.37 | 29.02 ± 0.52 | 27.43 ± 0.71 | 15.31 ± 0.36 | 28.14 ± 1.11 |
| | | 10 | 50.99 ± 0.62 | 34.06 ± 0.40 | 43.96 ± 1.37 | 16.51 ± 0.89 | 34.36 ± 0.35 |
| | | 50 | 56.81 ± 0.44 | 31.63 ± 0.55 | 45.94 ± 1.41 | 17.98 ± 1.06 | 30.14 ± 0.51 |
| | DSA | 1 | 27.76 ± 0.47 | 25.04 ± 0.77 | 25.59 ± 0.56 | 15.12 ± 0.65 | 23.70 ± 0.20 |
| | | 10 | 52.96 ± 0.41 | 34.49 ± 0.47 | 42.11 ± 0.56 | 16.10 ± 1.03 | 31.88 ± 0.35 |
| | | 50 | 60.28 ± 0.37 | 41.01 ± 0.36 | 49.52 ± 0.72 | 19.65 ± 1.16 | 43.30 ± 0.43 |
| | DM | 1 | 26.45 ± 0.39 | 10.02 ± 0.55 | 20.64 ± 0.47 | 14.09 ± 0.58 | 20.47 ± 0.46 |
| | | 10 | 47.64 ± 0.55 | 34.44 ± 0.30 | 38.21 ± 1.05 | 15.60 ± 1.51 | 34.37 ± 0.49 |
| | | 50 | 61.99 ± 0.33 | 40.49 ± 0.38 | 52.76 ± 0.44 | 21.67 ± 1.34 | 45.22 ± 0.37 |
| | KIP | 1 | 40.55 ± 1.34 | 26.31 ± 0.35 | 27.63 ± 1.06 | 14.16 ± 0.84 | 17.31 ± 1.63 |
| | | 10 | 47.23 ± 0.40 | 23.58 ± 0.38 | 38.82 ± 0.69 | 15.90 ± 0.21 | 15.85 ± 1.07 |
| | | 50 | 56.94 ± 0.38 | 25.25 ± 0.28 | 47.56 ± 0.76 | 18.44 ± 0.34 | 18.28 ± 0.64 |
| | MTT | 1 | 44.19 ± 1.18 | 10.40 ± 0.48 | 34.17 ± 1.41 | 13.40 ± 0.86 | 21.53 ± 0.44 |
| | | 10 | 63.66 ± 0.38 | 30.77 ± 0.60 | 45.22 ± 1.37 | 16.61 ± 1.37 | 33.58 ± 0.56 |
| | | 50 | 70.28 ± 0.61 | 38.45 ± 0.27 | 59.96 ± 0.72 | 20.90 ± 1.60 | 47.72 ± 0.57 |
| CIFAR-100 | Random | 1 | 5.30 ± 0.23 | 4.27 ± 0.09 | 4.36 ± 0.15 | 1.73 ± 0.12 | 4.45 ± 0.15 |
| | | 10 | 18.64 ± 0.25 | 10.20 ± 0.18 | 15.77 ± 0.24 | 5.19 ± 0.46 | 14.07 ± 0.21 |
| | | 50 | 34.66 ± 0.41 | 16.80 ± 0.31 | 30.23 ± 0.61 | 18.55 ± 1.29 | 26.90 ± 0.33 |
| | K-Center | 1 | 10.89 ± 0.17 | 7.96 ± 0.17 | 8.75 ± 0.43 | 2.22 ± 0.19 | 7.81 ± 0.13 |
| | | 10 | 25.04 ± 0.30 | 13.92 ± 0.20 | 22.18 ± 0.59 | 7.14 ± 0.79 | 17.98 ± 0.44 |
| | | 50 | 38.64 ± 0.43 | 19.32 ± 0.36 | 34.00 ± 0.51 | 21.25 ± 1.46 | 30.12 ± 0.65 |
| | DC | 1 | 13.66 ± 0.29 | 9.78 ± 0.27 | 9.71 ± 0.46 | 2.67 ± 0.16 | 9.27 ± 0.14 |
| | | 10 | 28.42 ± 0.29 | 12.36 ± 0.20 | 17.94 ± 0.59 | 5.28 ± 1.05 | 12.22 ± 0.17 |
| | | 50 | 30.56 ± 0.56 | 13.29 ± 0.30 | 17.64 ± 0.31 | 11.36 ± 0.95 | 17.51 ± 0.15 |
| | DSA | 1 | 13.73 ± 0.45 | 10.56 ± 0.22 | 9.95 ± 0.55 | 2.95 ± 0.44 | 9.48 ± 0.27 |
| | | 10 | 32.23 ± 0.35 | 16.17 ± 0.26 | 21.86 ± 0.43 | 5.45 ± 1.04 | 19.61 ± 0.15 |
| | | 50 | 43.13 ± 0.33 | 21.42 ± 0.31 | 34.34 ± 0.44 | 20.79 ± 1.76 | 31.89 ± 0.49 |
| | DM | 1 | 11.20 ± 0.27 | 8.17 ± 0.21 | 5.36 ± 0.31 | 2.11 ± 0.13 | 4.59 ± 0.26 |
| | | 10 | 29.23 ± 0.26 | 14.68 ± 0.18 | 18.72 ± 0.49 | 3.91 ± 0.73 | 17.06 ± 0.25 |
| | | 50 | 42.32 ± 0.37 | 20.14 ± 0.24 | 33.34 ± 0.40 | 17.29 ± 2.41 | 30.11 ± 0.25 |
| | KIP | 1 | 12.04 ± 0.15 | 5.55 ± 0.25 | 5.00 ± 0.38 | 1.95 ± 0.12 | 7.23 ± 0.34 |
| | | 10 | 29.04 ± 0.34 | 9.00 ± 0.21 | 20.99 ± 0.53 | 4.53 ± 0.18 | 12.05 ± 0.65 |
| | MTT | 1 | 22.3 ± 0.55 | 8.69 ± 0.33 | 13.32 ± 1.29 | 2.54 ± 0.11 | 4.65 ± 0.22 |
| | | 10 | 38.18 ± 0.42 | 14.35 ± 0.24 | 26.78 ± 0.58 | 5.74 ± 0.60 | 19.06 ± 0.31 |
| | | 50 | 46.32 ± 0.26 | 21.92 ± 0.27 | 41.08 ± 0.29 | 27.87 ± 1.99 | 32.93 ± 0.43 |
| TinyImageNet | Random | 1 | 1.65 ± 0.11 | 1.37 ± 0.08 | 1.27 ± 0.08 | 0.63 ± 0.08 | 1.71 ± 0.03 |
| | | 10 | 6.88 ± 0.25 | 3.12 ± 0.13 | 3.34 ± 0.16 | 1.01 ± 0.15 | 6.63 ± 0.21 |
| | | 50 | 18.62 ± 0.22 | 5.28 ± 0.2 | 10.35 ± 0.33 | 2.90 ± 0.40 | 16.87 ± 0.20 |
| | K-Center | 1 | 3.03 ± 0.12 | 2.53 ± 0.13 | 2.29 ± 0.10 | 0.82 ± 0.10 | 2.27 ± 0.02 |
| | | 10 | 11.38 ± 0.26 | 4.73 ± 0.08 | 5.46 ± 0.24 | 1.55 ± 0.21 | 9.92 ± 0.34 |
| | | 50 | 22.02 ± 0.40 | 5.99 ± 0.17 | 13.51 ± 0.34 | 3.91 ± 0.49 | 19.70 ± 0.41 |
| | DC | 1 | 5.27 ± 0.10 | 2.67 ± 0.17 | 3.17 ± 0.21 | 0.90 ± 0.14 | 2.00 ± 0.12 |
| | | 10 | 12.83 ± 0.14 | 4.12 ± 0.11 | 5.44 ± 0.21 | 1.24 ± 0.18 | 4.17 ± 0.10 |
| | | 50 | 12.66 ± 0.36 | 3.81 ± 0.17 | 7.05 ± 0.21 | 2.39 ± 0.21 | 5.22 ± 0.23 |
| | DSA | 1 | 5.67 ± 0.14 | 3.90 ± 0.16 | 3.20 ± 0.13 | 0.84 ± 0.12 | 3.17 ± 0.03 |
| | | 10 | 16.34 ± 0.21 | 6.31 ± 0.21 | 7.60 ± 0.36 | 1.90 ± 0.21 | 11.17 ± 0.15 |
| | | 50 | 25.31 ± 0.22 | 6.72 ± 0.20 | 13.36 ± 0.40 | 3.78 ± 0.56 | 19.87 ± 0.44 |
| | DM | 1 | 3.82 ± 0.21 | 3.11 ± 0.10 | 1.79 ± 0.17 | 0.89 ± 0.07 | 3.21 ± 0.07 |
| | | 10 | 13.51 ± 0.31 | 4.24 ± 0.13 | 3.57 ± 0.20 | 1.06 ± 0.19 | 7.46 ± 0.20 |
| | | 50 | 22.76 ± 0.28 | 5.74 ± 0.27 | 11.07 ± 0.39 | 3.33 ± 0.46 | 18.88 ± 0.36 |
| | MTT | 1 | 8.27 ± 0.36 | 3.33 ± 0.13 | 4.45 ± 0.23 | 0.95 ± 0.09 | 2.25 ± 0.07 |
| | | 10 | 20.11 ± 0.16 | 4.59 ± 0.20 | 10.16 ± 0.43 | 1.74 ± 0.21 | 11.14 ± 0.24 |
| | | 50 | 28.16 ± 0.45 | 6.04 ± 0.20 | 20.65 ± 0.44 | 4.10 ± 0.52 | 21.43 ± 0.25 |

Table 6: Test accuracy of condensation methods under different IPCs. All numbers are recorded on ConvNet and CIFAR-10 dataset with DSA augmentation.

| IPC | Method | | | | |
| --- | --- | --- | --- | --- | --- |
| | Random | K-Center | DC | DSA | DM |
| 1 | $15.40 \pm 0.28$ | $25.16 \pm 0.45$ | $29.34 \pm 0.37$ | $27.76 \pm 0.47$ | $26.45 \pm 0.39$ |
| 10 | $31.00 \pm 0.48$ | $41.49 \pm 0.73$ | $50.99 \pm 0.62$ | $52.96 \pm 0.41$ | $47.64 \pm 0.55$ |
| 50 | $50.55 \pm 0.32$ | $56.00 \pm 0.29$ | $56.81 \pm 0.44$ | $60.28 \pm 0.37$ | $61.99 \pm 0.33$ |
| 100 | $57.89 \pm 0.57$ | $62.18 \pm 0.18$ | $65.70 \pm 0.44$ | $66.18 \pm 0.30$ | $65.12 \pm 0.40$ |
| 200 | $64.70 \pm 0.44$ | $67.25 \pm 0.48$ | $68.41 \pm 0.45$ | $69.49 \pm 0.13$ | $69.15 \pm 0.17$ |
| 300 | $68.52 \pm 0.29$ | $70.12 \pm 0.09$ | $69.42 \pm 0.45$ | $71.46 \pm 0.27$ | $69.36 \pm 0.35$ |
| 400 | $70.28 \pm 0.39$ | $71.88 \pm 0.34$ | $70.86 \pm 0.42$ | $72.22 \pm 0.48$ | $72.61 \pm 0.54$ |
| 500 | $73.19 \pm 0.29$ | $74.31 \pm 0.33$ | $72.05 \pm 0.35$ | $73.62 \pm 0.28$ | $75.09 \pm 0.26$ |
| 600 | $74.00 \pm 0.29$ | $75.98 \pm 0.19$ | $72.84 \pm 0.40$ | $74.99 \pm 0.21$ | $76.07 \pm 0.20$ |
| 700 | $75.29 \pm 0.25$ | $76.74 \pm 0.34$ | $73.70 \pm 0.31$ | $76.07 \pm 0.26$ | $76.78 \pm 0.27$ |
| 800 | $75.52 \pm 0.10$ | $76.94 \pm 0.21$ | $74.80 \pm 0.29$ | $76.76 \pm 0.27$ | $77.41 \pm 0.19$ |
| 900 | $77.44 \pm 0.45$ | $78.21 \pm 0.20$ | $75.26 \pm 0.34$ | $77.77 \pm 0.44$ | $78.33 \pm 0.41$ |
| 1000 | $78.38 \pm 0.20$ | $79.47 \pm 0.29$ | $76.62 \pm 0.32$ | $78.68 \pm 0.25$ | $78.83 \pm 0.05$ |

Table 7: Test accuracy of different condensation methods with SGD and Adam optimizer on CIFAR-10. All results are evaluated with DSA augmentation.

| Optimizer | IPC | Random | K-Center | DC | DSA | DM | KIP | MTT |
| --- | --- | --- | --- | --- | --- | --- | --- | --- |
| SGD | 1 | $15.40 \pm 0.28$ | $25.16 \pm 0.45$ | $29.34 \pm 0.37$ | $27.76 \pm 0.47$ | $26.45 \pm 0.39$ | $40.55 \pm 1.34$ | $44.19 \pm 1.18$ |
| | 10 | $31.00 \pm 0.48$ | $41.49 \pm 0.73$ | $50.99 \pm 0.62$ | $52.96 \pm 0.41$ | $47.64 \pm 0.55$ | $47.23 \pm 0.40$ | $63.66 \pm 0.38$ |
| | 50 | $50.55 \pm 0.32$ | $56.00 \pm 0.29$ | $56.81 \pm 0.44$ | $60.28 \pm 0.37$ | $61.99 \pm 0.33$ | $56.94 \pm 0.38$ | $70.28 \pm 0.61$ |
| Adam | 1 | $15.39 \pm 0.38$ | $25.40 \pm 1.21$ | $28.76 \pm 0.71$ | $27.83 \pm 1.27$ | $26.22 \pm 0.40$ | $30.56 \pm 3.07$ | $46.62 \pm 1.39$ |
| | 10 | $32.71 \pm 0.92$ | $43.26 \pm 0.41$ | $52.01 \pm 0.62$ | $51.17 \pm 0.91$ | $48.64 \pm 1.02$ | $44.20 \pm 0.62$ | $64.94 \pm 0.61$ |
| | 50 | $49.75 \pm 1.10$ | $54.82 \pm 1.23$ | $54.75 \pm 1.52$ | $58.73 \pm 0.96$ | $61.13 \pm 0.92$ | $55.08 \pm 1.88$ | $71.82 \pm 0.52$ |

Table 8: Training time of different condensation methods. All results for synthesis based methods are acquired by running 100 iterations and all results for K-Center are acquired by performing 300 iterations on CIFAR-10 with IPC 1, 10, 50. The average result and standard deviation per iteration are reported for each IPC

| Method | run time(sec) | | | GPU memory(MB) | | |
| --- | --- | --- | --- | --- | --- | --- |
| | 1 | 10 | 50 | 1 | 10 | 50 |
| K-Center | $0.0012 \pm 0.00$ | $0.015 \pm 0.00$ | $0.05 \pm 0.00$ | 3575 | 3575 | 3575 |
| DC | $0.16 \pm 0.01$ | $3.31 \pm 0.02$ | $15.74 \pm 0.10$ | 3515 | 3621 | 4527 |
| DSA | $0.22 \pm 0.02$ | $4.47 \pm 0.12$ | $20.13 \pm 0.58$ | 3513 | 3639 | 4539 |
| DM | $0.08 \pm 0.02$ | $0.08 \pm 0.02$ | $0.08 \pm 0.02$ | 3323 | 3455 | 3605 |
| MTT | $0.36 \pm 0.23$ | $0.40 \pm 0.20$ | OOM | 2711 | 8049 | OOM |

Note: different methods need different iterations to converge, this table shows the run time and memory needed per iteration which reflect the bottleneck of the algorithm.

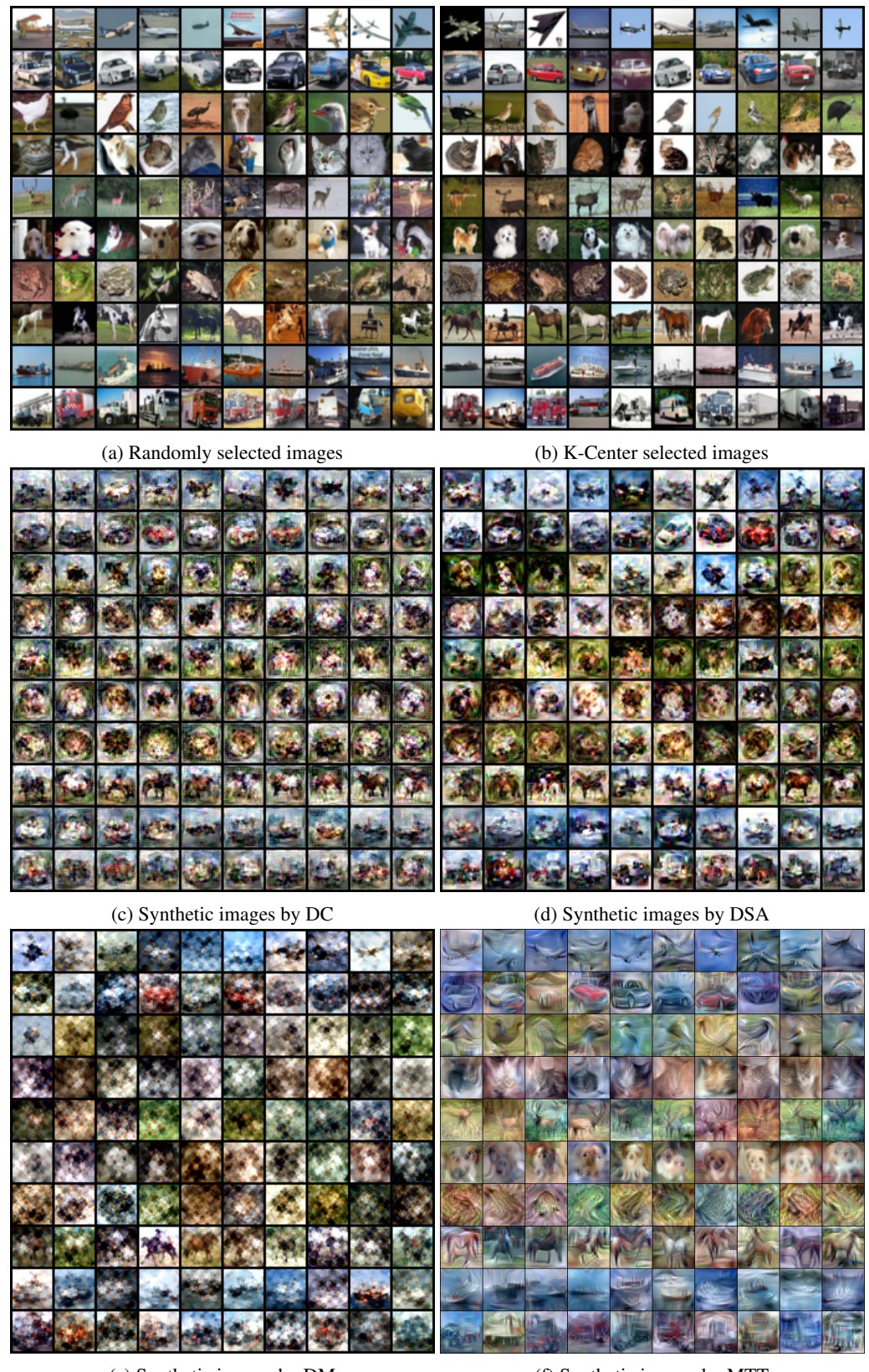

(a) Randomly selected images

(b) K-Center selected images

(c) Synthetic images by DC

(d) Synthetic images by DSA

(e) Synthetic images by DM

(f) Synthetic images by MTT

Figure 8: Real images vs synthetic images