# OpenReview forum: "DC-BENCH: Dataset Condensation Benchmark"
_NeurIPS.cc/2022/Track/Datasets_and_Benchmarks — NeurIPS 2022 Datasets and Benchmarks _

### Official Review · Reviewer_wYHF · 2022-07-18
**A good contribution for the DC field**

**Rating:** 8
**Confidence:** 2
**Correctness:** The claims seem correct.
**Clarity:** There are many mistakes that should b…

**Strengths:**

The paper addresses an important problem in ML and thanks to benchmarking is able to underlines several shortcomings of the field as well as isolate the effect of various processes.

**Weaknesses:**

There are many errors that make the reading a difficult experience, and no discussion of how the benchmark is going to be updated with new models.

**Additional Feedback:**

The combination of data selection and data synthesis is a bit confusing because some of the methods do not need a set of selected data, such as [51].

**Documentation:**

The benchmark is well documented and while I did not attempt to run it the github seems clear and well organized.

**Ethics:**

There are no ethical problems.

**Relation To Prior Work:**

Relation to prior work is clearly discussed, with what the paper brings to the table highlighted.

**Summary And Contributions:**

The authors propose a dataset condensation benchmark for images, including several architectures, the effect of data augmentation, and the performance on NAS task. They compare four methods: DC, DSA, DM, and TM.

---

### Official Review · Reviewer_5Lwd · 2022-07-20
**Valuable insights but benchmark requires further documentation and restructuring extensibility.**

**Rating:** 5
**Confidence:** 4

**Strengths:**

- Extensive benchmarking with 3 datasets, 4 augmentation strategies, 4 different model architectures, 4 data augmentation, and 2 data selection methods.
- Several insights on the scalability and generalizability of existing data condensation methods.
- Evaluation also compares methods on a wider range of compression ratios, with larger IPCs to evaluate in use-cases with less stringent compression requirements.

**Weaknesses:**

There are several parts that require further explanation, and some of the experimental setup choices are not justified (more details below). My main concern is about the ease of use and extensibility of the benchmark. In other words, it is unclear how easy it is to include new methods being developed. The GitHub repository does not contain any such documentation (and cannot seem to find such a class interface).

In general, documentation and project structure is not very clear. When trying to find the code for each method, [distribution_matching.py](https://github.com/justincui03/dc_benchmark/blob/main/methods/distribution_matching/distribution_matching.py) seems to be an empty python file and [mtt-distillation](https://github.com/justincui03/dc_benchmark/tree/main/methods/mtt-distillation) and others seem to be copies of full repositories from each paper. Would be good to consider a common structure across methods, and to create a well-documented API with code releases and GitHub Docs (example: https://github.com/openai/gym)

Same comments hold for extending the toolkit to other datasets, e.g., domain-specific ones that a future user would like to work on, as well as new data augmentation methods, models, or downstream tasks. Are there instructions on how to do so? Would be good for the paper and/or repository to cover such details. Does the open-sourced code contain **unit tests** or any other form of code verification to ensure correctness and fix any future bug fixes?

**Additional Feedback:**

To confirm, are all results reported before section 3.6 with randomly initialized synthetic images?

In general, the reviewer appreciates the extensive experimental analysis of data condensation methods. The toolkit would have been a great resource if it contained an easily extendable API for new methods to evaluate contributions easily. Benchmarking on larger datasets and including training times as another comparison dimension would further strengthen this work.





**Clarity:**

The paper is generally well written, but with various typos that can be easily fixed with careful review, e.g. line 23 ("contemporary datases"), line 46 ("the relative performance of condensation methods on real-world downstream applications *are* also rarely discussed"), line 237 ("Follow*ing* previous works") and many more.

**Correctness:**

For K-Center operating on the raw input space (lines 215-222), can the authors confirm that this is indeed true? Based on [51]: " For Herding and K-Center, we use models trained on the whole dataset to extract features, ...", and thus it seems that [51] reports results with K-Center and representations extracted from trained models (?).

In terms of other experimental details:
1) Why focus on NAS only as a downstream task that should benefit from data condensation? Why not consider Continual Learning as well, or other applications?
2) To test scalability, it would be good to also consider evaluation on larger datasets such as ImageNet, or at least a subset with larger class sizes ( > 500 classes).
3) The motivation in lines 199-202 on other data selection methods, and why these are not included seems unclear to me. Can the authors further explain their reasoning, e.g., what is the main difference that prevents comparison?
4) Are results reported over multiple experimental trials? Especially for Fig. 4, it would make sense to show variance.
5) In Appendix, there is a statement in line 585 "We are not able to get the performance of TM under the computation resource limit we set." Please add more details on what this limit was (1000 training epochs? training time? GPU memory?)
6) Lines 251-252, what is the reasoning behind controlling data augmentation only for the evaluation step? If two different data condensation methods apply different augmentations, then couldn't the choice of augmentation substantially alter performance? Why not control all such factors? What does applying the right data augmentation during evaluation mean? What is considered the best data augmentation, is this decided by downstream performance or compression ratio?
7) Please further explain Table 4 caption, "The rank correlation of original dataset is lower than 1.0 because we use a small architecture and perform ranking based on validation set.", in particular, which of the two contributes more to the rank correlation, the smaller architecture or the ranking on the validation set?

**Documentation:**

Documentation on Github or in the paper is fairly limited. Suggestions can be found in the Weaknesses section.

**Relation To Prior Work:**

The contribution of the proposed benchmark toolkit is clear.

**Summary And Contributions:**

This paper introduces a data condensation/distillation benchmark under different model architectures, compression ratios, and datasets, expanding comparisons of prior work to demonstrate how such factors can affect the end performance. The evaluation also considers a neural architecture search scenario, and two data selection mechanisms (both as baselines and as initialization strategies).

---

> ### Author Response · Authors · 2022-08-19
> **Response to reviewer 5Lwd P3**
>
> ## Correctness
> **Q5: Are results reported over multiple experimental trials? Especially for Fig. 4, it would make sense to show variance.**
>
> R5: Thank you for the suggestion.  Yes, the results are over multiple trials and we added variance to the graph.
>
> **Q6: In Appendix, there is a statement in line 585 "We are not able to get the performance of TM under the computation resource limit we set." Please add more details on what this limit was (1000 training epochs? training time? GPU memory?)**
>
> R6: Thank you, we added the reasons(out of memory) to make it more clear. They are also indicated in Table 8.
>
> **Q7: Lines 251-252, what is the reasoning behind controlling data augmentation only for the evaluation step? If two different data condensation methods apply different augmentations, then couldn't the choice of augmentation substantially alter performance? Why not control all such factors? What does applying the right data augmentation during evaluation mean? What is considered the best data augmentation, is this decided by downstream performance or compression ratio?**
>
> R7: The first reason we conduct this experiment is because some method like DSA claims augmentation is not useful during evaluation if the augmentation is already used during data synthesis phase. Our experiments prove that augmentation is very useful for synthesized dataset even if only applied in the evaluation phase. It was missing in some previous works. Therefore we did a fair evaluation by applying it to all methods.
>
> Our second motivation is that, as an end user, we want to use the synthetic dataset just like regular datasets without knowing which augmentation was used to generate them. And end users are free to choose whichever augmentation that best suits their tasks. That’s also the reason we report average performance and best performance to show that end users have to decide which(the right) augmentation best fits their needs.
>
> **Q8: Please further explain Table 4 caption, "The rank correlation of original dataset is lower than 1.0 because we use a small architecture and perform ranking based on validation set.", in particular, which of the two contributes more to the rank correlation, the smaller architecture or the ranking on the validation set?**
>
> R8: This is a great question. We ran an experiment and found out that the spearman’s ranking correlation between validation set and test accuracy is 0.9792. The spearman’s ranking correlation using validation set and smaller network is 0.7487 as reported in our paper. Therefore, the smaller architecture contributes more to ranking correlation.

---

> > ### Comment · Reviewer_5Lwd · 2022-08-29
> > **Thank you for the rebuttal**
> >
> > Thank you to the authors for their detailed responses. In light of the revisions, I am slightly raising my score. However, the instructions for adding new methods require more details. One can *kind of* understand based on the README files how to include a new dataset or a new method, but polishing both documents and also adding a class interface that users can easily populate would be strongly recommended. Adding unit tests would also be a nice plus for verification. There remain other concerns raised by reviewers; my comments focus on the usability / ease of use of the proposed benchmark.

---

> > > ### Author Response · Authors · 2022-08-29
> > > **Thank you for your reply!**
> > >
> > > Thank you for reviewing our responses and raising the evaluation. We really appreciate it.
> > >
> > > **Q1: However, the instructions for adding new methods require more details. One can kind of understand based on the README files how to include a new dataset or a new method, but polishing both documents and also adding a class interface that users can easily populate would be strongly recommended. Adding unit tests would also be a nice plus for verification. There remain other concerns raised by reviewers; my comments focus on the usability / ease of use of the proposed benchmark.**
> > >
> > > R1: Thank you for your feedback. Dataset synthesis related techniques are evolving very fast, and it is difficult to factor the benchmark at the same level as an established python library. As the first open sourced benchmark in the field, we have included examples in the instructions in the codebase. We will polish the document more and refactor the code as you suggested.
> > >
> > > We are committed to keep updating and maintaining the benchmark. We are also actively engaging with the community to improve the usability of the benchmark. For example, we have included KIP(** a new method **)  and ViT(** a new model) into our work. We are also currently working to include the newly published work: Frepo(https://arxiv.org/abs/2206.00719) and Efficient Synthetic-Data Parameterization(https://proceedings.mlr.press/v162/kim22c.html) while improving our interfaces. We are also adding more use cases such as Continual Learning and Membership Inference Defense(used in new Efficient Synthetic-Data Parameterization work). We are confident that the benchmark usability will be greatly improved while adopting more use cases from the community.
> > >
> > > In terms of unit tests, it’s also on our road map. We will keep refactoring our interfaces to make them easier to use and keep updating our benchmark/leaderboard with newest methods and applications to keep them up to date. We sincerely hope this contributes to the community and helps build a unified benchmark for the field.

---

> ### Author Response · Authors · 2022-08-19
> **Response to reviewer 5Lwd P2**
>
> ## Correctness
>
> **Q1: For K-Center operating on the raw input space (lines 215-222), can the authors confirm that this is indeed true? Based on [51]: " For Herding and K-Center, we use models trained on the whole dataset to extract features, ...", and thus it seems that [51] reports results with K-Center and representations extracted from trained models (?).**
>
> R1: Thank you very much for pointing it out. We have removed this claim for better clarification in our revised version. The reason we had this suspicion is that our results for K-Center are much better than previous reported ones for the same datasets. We think it is probably due to overtraining of the feature extractor model for too many epochs — in experiment, we observed that training for 1 epoch achieves best result, otherwise the features will be too close and cause downgraded performance.
>
> **Q2: Why focus on NAS only as a downstream task that should benefit from data condensation? Why not consider Continual Learning as well, or other applications?**
>
> R2:Since dataset condensation aims at accelerating model training, it is essential to study how the condensed dataset effects model training. From this perspective, we deem NAS as the principled systematic method for evaluation.
>
> Also NAS evaluation was performed in some of the works. However, different data condensation papers apply different search space and search methods for NAS which makes it hard to compare the results between different papers. In the NAS community there are already standardized benchmarks such as NAS-Bench-201. So we think it is clear that the condensation community should also compare their methods on this benchmark to have a fair comparison between different methods. We want to encourage the community to use standardized reproducible benchmarks to evaluate it.
>
> We agree that continual learning is another application of data condensation. However, since the algorithms for continual learning are still evolving quickly in recent years and we need more time to investigate what's the standard benchmark and method used in continual learning. Therefore, we plan to add continual learning as well as other tasks in the future.
>
> **Q3: To test scalability, it would be good to also consider evaluation on larger datasets such as ImageNet, or at least a subset with larger class sizes ( > 500 classes).**
>
> Q3: That’s a really good point. It aligns with our primary goal very well. We chose CIFAR10, CIFAR100 and TinyImagenet for the same scalability reason. Previously, lots of them used datasets including MNIST, FashionMNIST which are too small to reveal the true performances such as scalability and efficiency. To the best of our knowledge, none of the published existing condensation methods works with datasets larger than TinyImagenet with higher resolutions. With this benchmark, we try to encourage the research community to shift focus from small datasets to medium and large datasets exactly as you suggested.We hope our contribution can help accelerate the process.
>
> **Q4: The motivation in lines 199-202 on other data selection methods, and why these are not included seems unclear to me. Can the authors further explain their reasoning, e.g., what is the main difference that prevents comparison?**
>
> R4: Thank you for pointing this out. Our primary goal for DC-Bench is to guide the research direction of dataset synthesis methods. Selection based methods that have been studied for decades with theoretical analysis are not the focus of our current benchmark. We chose to just include random selection which is used in almost all dataset synthesis methods as the basic baseline and K-Center which is also a strong core-set selection baseline.
>
> As also suggested by other reviewers, in order not to confuse readers, we will just focus on data synthesis methods. This should make our benchmark’s focus more clear.

---

> ### Author Response · Authors · 2022-08-19
> **Response to reviewer 5Lwd P1**
>
> **Q1. My main concern is about the ease of use and extensibility of the benchmark. In other words, it is unclear how easy it is to include new methods being developed. The GitHub repository does not contain any such documentation**
>
> R1. Thank you for pointing out that extensibility is key to the benchmark. We have updated github with detailed documentations on how to integrate new methods and dataset. At the same time, we have added a way for the community to submit their new SOTA results through our leaderboard: https://dc-bench.github.io/ . We will keep updating the benchmark and leaderboard constantly to keep them up to date. Since our github is online, we are contacted by several groups and for example the KIP group is working with us on adding their method to the benchmark.
>
> **Q2. In general, documentation and project structure is not very clear. When trying to find the code for each method, distribution_matching.py seems to be an empty python file and mtt-distillation and others seem to be copies of full repositories from each paper.**
>
> R2. Thank you for your suggestion. Based on your feedback, we have modified our project structure and added more documentation on how to run experiments, integrate new methods/dataset and how to reproduce SOTA. The empty file is due to an accidental push. We have removed it from our codebase. Also for the copies of other repositories, we added them for easy reproduction of SOTA methods, e.g for mtt-distrition(https://github.com/justincui03/dc_benchmark/blob/main/methods/tm/readme.md), some of the hyperparameters cannot even by found on the original author’s codebase before. We included all of them for the convenience of the community.
>
> **Q3. Same comments hold for extending the toolkit to other datasets,  e.g., domain-specific ones that a future user would like to work on, as well as new data augmentation methods, models, or downstream tasks. Are there instructions on how to do so? Would be good for the paper and/or repository to cover such details. Does the open-sourced code contain unit tests or any other form of code verification to ensure correctness and fix any future bug fixes**
>
> R3. Thank you for your suggestion. We have updated our github page with instructions on how to integrate new methods, dataset and models. Our codebase is designed to be modularized. Therefore, it should be easy to extend(we have provided detailed instructions on how to integrate). In terms of downstream tasks, they usually require us to integrate a whole new codebase. We can start by following the way we did for NAS where we incorporated the NAS codebase and provided the scripts to run evaluation.
>
> In terms of unit tests, we don’t have any so far. But we have manually verified the components of our benchmark. E.g. We evaluated the models used in our benchmark on real datasets with various augmentations and verified the performances.  But we agree with you that adding unit tests is a good idea to ensure correctness of codes. We will explore different options and add them into our benchmark in future updates. Also we will be constantly updating our benchmarks and leaderboard including adding new SOTA methods and bug fixes.

---

### Official Review · Reviewer_SxsG · 2022-07-26
**A very specific work that would benefit from a broader literature embedding and assessment**

**Rating:** 5
**Confidence:** 4

**Strengths:**

* The paper is well structured, easy to read and to follow.
* The four considered dataset condensation methods are very recent and intuitively promising. In particular there is an appeal that these may be better than traditional core sets (which may or may not be the case, as some of the empirical investigation has shown here)
* The idea to establish a leaderboard that is being updates is nice. Perhaps not primarily for the reason to be “state of the art”, but given that there are several dimensions of evaluation, more so for comparison purposes in order to relate methods. (Maybe this could also be emphasized, as the present narrative seems to favor “stat of the art”)
* The neural architecture search investigation is interesting, because it is an experiment that many may want to conduct, albeit will find, as shown, that the correlations are negligible. These types of empirical insights may be useful to the community.

**Weaknesses:**


* The choice of methods and taxonomy seems overly narrow. Already when reading lines 28-30 it does not become clear why this particular taxonomy is meaningful. Most importantly, it is unclear to me why “dataset condensation” is framed as a novel and “emerging promising direction” (lines 26). This may indeed be true for the particular idea in the four synthesis method, but only specifically with respect to combining multiple data points/classes into a single instance. If we think of data subsets, core sets have enjoyed rich theoretical investigation and a long history of empirical methods for their selection. It is somewhat surprising to not even have core sets be mentioned, and two oddly specific methods (herding and k-center) be mentioned as some “recent” advocates of data subset selection.
* My biggest concern, stemming directly from above argument on the lack of mention to various prior works, is that the framing makes it seem like much of the work is particularly novel and insightful, even if it is at least in parts well known. More precisely, the notion that “kmeans-embed” is some novel proposed method to select data subsets is not only very stretched, but just plain wrong. Core sets have a long history and the particular suggested approach is far from novel: see e.g. Tsang et al 2005 for SVMs, core sets for k-means and k-medians in Har-Peled & Kushal 2005, or concretely with respect to the novelty of the “evaluating kmeans in the embedding” the robust k-center based on l2 distances in activations of a deep convolutional layer in e.g. the core set approach of Sener & Savarese ICLR 2018. Overall, the key difference is that these methods not only solve the data subset selection problem, they actually do so with various formal guarantees to which degree they work. It’s a very common paradigm that finds applications from active learning to continual learning.  In my humble opinion, this particular paper under review would actually do itself a favor to remove the kmeans embed part and its advocacy for novelty and stick to the investigation of the condensation methods, rather than core sets. On the flip side, if the paper does in fact want to draw an analogue to core sets, it should do so rigorously and consider more than one method (in addition to random sampling) towards a more exhaustive and faithful comparison.
* The data augmentation part is interesting in principle, but it is unclear to me what precise insights are supposed to be drawn from the benchmark as a practitioner. Rather then proposing and conducting a very systematic ablation study on the influence of various types of data augmentation in data condensation creation and later training of a derivate model, the shown study conflates various augmentation factors. It unfortunately becomes completely opaque in terms of which factors contribute when, particularly as some augmentation techniques are overlapping and superpositions.
* Whereas some experiments seem to have a motivation, others feel more ad-hoc. For instance, what was the hypothesis and expectation behind section 3.3? It is well-known and trivial (from core sets and intuition) that larger amounts of images per class will boost accuracy (figure 2). In core sets, what is then typically investigated is the quality of the selection mechanism, in particular for small sample scenarios. Here however, the dataset condensation methods are all biased by an already picked set of randomly subsampled data instances.
* It would be great to see standard deviations/any measures of statistical deviations across randomly seeded experimental repetitions in any of the experimental result figures. Given that all of the techniques are extremely stochastic, I suspect that the addition of standard deviations would show that there is virtually fairly little difference between the methods in many cases, in particular figure 1 + 2.

**Additional Feedback:**

The paper tackles an overall interesting problem. A more thorough discussion of related work and analysis beyond a small set of specific methods could elevate it significantly.

**Clarity:**

See above point for correctness, also referring in parts to clarity, with respect to the use of specific sets of weights and model ensembles.
Similarly, see below point for documentation, referencing some aspects where clarity would increase through extended documentation.

**Correctness:**

The transferability and architecture experiments are also a bit unclear to me with respect to the supposed take-away and whether they are “correct” (in the sense of following the assumptions correctly)

In principle I understood its motivation, but was then puzzled when the text implied (please correct me if I am wrong) that the baseline architecture with which the condensed examples are generated is always the same. Why is this a reasonable assumption for some of the condensation methods, e.g. the ones that are based on very specific trajectories of a particular model, their specific parameters and statistics? The discussion I was lacking in the paper is that even the original papers, like the DC with gradient matching paper, show that in order for condensed examples to be generally useful, one either needs to use a large ensemble of methods to generate them under diverse sets of (potentially random) weights, or suffer from major losses when assuming that the precisely condensed solution is generic. To the best of my knowledge, the trajectory matching paper follows a similar assumption, and uses a large ensemble set of models as the basis. It is unclear to me which of these assumptions were dropped and why this would be reasonable in experimentation.


**Documentation:**

The documentation could be substantially improved. First, a short mathematical or algorithmically summary of the benchmarked methods would be appreciated (in supplementary). Second, many of the training details are only mentioned in passing. In particular, the use of data augmentation in terms of whether a method uses it for condensation and then later model training on the condensed examples, or just one of the two, does not become fully clear (and why this is reasonable). The appropriate section in the supplementary material is too short and only details a very rough set-up in 9 lines.

**Ethics:**

There are no ethical concerns.

**Relation To Prior Work:**

As illustrated in the weakness section, prior work is not well discussed. The paper relates sufficiently to 4 specific examples of dataset condensation, but leaves aside a rich and decades old literature surrounding core set selection and heuristic procedures to extract useful data subsets. As a suggested benchmark, this is particularly worrying because 3 out of the 4 methods are from a single author team, rendering the assessment and overall discussion in context of the overall literature unfortunately narrow.

**Summary And Contributions:**

The paper proposes a benchmark for dataset condensation techniques. More specifically, it aims to identify various factors to be analyzed when investigation dataset condensation, revolving primarily about numbers of instances, choice of architecture, and data augmentation methods. Together with the additional aspect of transferability, these factors are then investigated empirically for the CIFAR and tinyImageNet datasets on the basis of four recently proposed dataset condensation techniques and two straightforward approaches to select original data instances to store.

Post rebuttal update:
As indicated in below response, parts of my concerns have been addressed, whereas decisions taken to (not) address others lead me to raise my rating while still leaning towards rejecting the paper in the revised form.

---

> ### Author Response · Authors · 2022-08-19
> **Response to reviewer SxsG P4**
>
> **Q7: As a suggested benchmark, this is particularly worrying because 3 out of the 4 methods are from a single author team, rendering the assessment and overall discussion in context of the overall literature unfortunately narrow**
>
> R7: Thank you for the feedback. At the time of writing, most dataset synthesis methods are indeed from the same group; however, these are also representative ones in the field. Each of them introduced a new technique that became popular in the community. For example, DC introduced the concept of gradient matching; DSA introduced differentiable augmentation; DM introduced the concept of embedding matching; TM introduced parameter matching. We will keep updating our benchmark to keep it up to date. We also created a portal in our leaderboard (https://dc-bench.github.io/) to allow future users to submit their new methods. Since the leaderboard is published online, we have received several requests for adding new and latest benchmarks.  For example, we have included KIP[cite?] upon the request from the authors. We hope our work will help accelerate the development of the new condensation methods.
>
> **Q8:The documentation could be substantially improved. First, a short mathematical or algorithmically summary of the benchmarked methods would be appreciated (in supplementary). Second, many of the training details are only mentioned in passing. In particular, the use of data augmentation in terms of whether a method uses it for condensation and then later model training on the condensed examples, or just one of the two, does not become fully clear (and why this is reasonable). The appropriate section in the supplementary material is too short and only details a very rough set-up in 9 lines.**
>
> R8:Thank you for your feedback. We made several changes according to your suggestions. In order to make our structure more clear, we separated related work into a standalone section and added mathematical summaries for the included methods. Hope this can help readers better understand the methods included in our benchmark.
>
> Regarding the augmentation. For model training, we stick to the settings the authors suggested in their original work. However, augmentation in training is not our focus. The authors can choose any augmentations to generate synthetic images. We treat the data generation process as a blackbox. When it comes to users who want to utilize the synthetic images for downstream tasks, they can choose the type of augmentation that’s tailored to their tasks. From a user’s point of view, we think this is more practical.
>
> We rephrase the paragraph in the experiment setup section to include these as well to make it more clear.
>
> **Q9: The paper tackles an overall interesting problem. A more thorough discussion of related work and analysis beyond a small set of specific methods could elevate it significantly.**
>
> R9: Thank you for your suggestion. We have separated related work with short mathematical summaries into a standalone section to make it easier to read. As the first benchmark in this field, at the time of writing, the majority of methods are indeed from the same author group. Our work has covered most of the newly proposed methods in the field and we hope our benchmark can facilitate the community to develop more methods.
>
> Also thanks for your great suggestion for including core set methods that have been well studied. We will be constantly updating our benchmarks and will consider adding them in our future benchmark version. So far we have received requests from the community to add their methods into our benchmark. We will keep working on it.

---

> > ### Comment · Reviewer_SxsG · 2022-08-29
> > **Feedback to the four response parts**
> >
> > I thank the authors for the detailed responses to my concerns.
> >
> > In light of the fact that some improvements have been made, I am slightly raising my score from a clear reject to a less strong one.
> > Unfortunately, I will still remain at a rating that leans towards rejection in the present form as I seem to disagree with the route of partial revisions that have been chosen in an attempt to improve the paper.
> >
> > More specifically, the parts that I think do improve the paper are the addition of the key take aways, even if they are at times rather shallow and potentially misleading in their simplistic form. The addition of error bars is further appreciated. The addition of the kernel ridge regression method, as advised by another reviewer, is also noted. The removal of the novelty claim and rewordings also make the paper much less misleading. For these reasons, I am raising my rating to a weak reject one.
> >
> > However, the way that data instance selection and core sets are treated is still insufficient. The fact that the novelty claim has been removed and rephrased to k-center does not make up for the fact that the discussion is missing. It is a good first step, but the appropriate relation to other works is still missing. Perhaps it is now even more peculiar than before, that core sets methods are first mentioned extremely briefly in the related work section and denoted to be out of scope, to then later find a claim that the used k-center
> > Here, I strongly disagree with the sentiment that "the results in table 1 outperform all coreset methods", which is anyhow peculiar now that coresets have been deemed out-of-scope and related works have not been discussed properly.
> >
> > Similarly, I do not think the data augmentation concern is as easily fixable as adding a simple statement that data augmentation is treated from an "end user perspective". I understand that an end user may care primarily about the final results, but I believe it is a fallacy to assume that an end user would not also be interested in a more principled study of data augmentation, where factors of variation are more clearly separated and not entangled in uninterpretable ways. Perhaps this is a matter of opinion after all, but I would expect benchmarks and dataset studies to be more systematic and principled here.
> >
> > Overall, I am thus unfortunately still not convinced to accept the benchmark in its present form. I believe a more principled analysis of the data augmentation parts together with a much more nuanced discussion of related work (e.g. core sets) and respective key take-aways are still required.

---

> > > ### Author Response · Authors · 2022-08-29
> > > **Thank you for your reply**
> > >
> > > We thank the reviewer for going through our responses and raising the evaluation as a confirmation of our updates. Please kindly see our response below.
> > >
> > > **Q1: However, the way that data instance selection and core sets are treated is still insufficient. The fact that the novelty claim has been removed and rephrased to k-center does not make up for the fact that the discussion is missing. It is a good first step, but the appropriate relation to other works is still missing. Perhaps it is now even more peculiar than before, that core sets methods are first mentioned extremely briefly in the related work section and denoted to be out of scope, to then later find a claim that the used k-center
> > > Here, I strongly disagree with the sentiment that "the results in table 1 outperform all coreset methods", which is anyhow peculiar now that coresets have been deemed out-of-scope and related works have not been discussed properly.**
> > >
> > > R1: We thank the reviewer for the suggestion and the effort in reviewing our work.
> > > We would like to first re-emphasize our full claim: “Our results in Table 1 show that it outperforms all coreset methods **used in [54, 53, 55, 4] under almost all settings**”. The bolded citations seem to be missing in your quote. We are only comparing our version of K-Center to the coreset methods[Random, Herding, K-Center, Forgetting] frequently used in recent dataset synthesis work.
> > >
> > > Our original intention was to use the set of coreset selection methods included in prior condensation works [dc, dsa, dm,tm,kip] only as baselines to benchmark the condensation methods [herding, K-Center, Forgetting]. Yet we found that most of these baselines implemented and reported by prior condensation papers perform worse than random selection, which we believe to largely underestimate the strength of these methods. We reimplemented K-Center and observed greatly improved performance. We therefore use our K-Center and random selection to baseline the condensation methods.
> > >
> > > As you mentioned, there is a rich and decades old literature of coreset methods that are worth investigating. Since we focus on recently developed condensation methods, we believe coreset methods **deserve a separate benchmark themselves**. We have started looking into the domain of coreset methods and will also share future results with the community.
> > >
> > > **Q2: Similarly, I do not think the data augmentation concern is as easily fixable as adding a simple statement that data augmentation is treated from an "end user perspective". I understand that an end user may care primarily about the final results, but I believe it is a fallacy to assume that an end user would not also be interested in a more principled study of data augmentation, where factors of variation are more clearly separated and not entangled in uninterpretable ways. Perhaps this is a matter of opinion after all, but I would expect benchmarks and dataset studies to be more systematic and principled here.**
> > >
> > > R2:
> > > Thank you for your feedback, our benchmark focuses on properly evaluation of the generated datasets using satete-of-the-art augmentations. As a result, we are viewing data augmentation from the end-user perspective, i.e. whether and how adding different augmentation during the evaluation of the generated datasets affects the performance of different condensed methods.
> > >
> > > Our primary goal is to encourage the community to develop generic synthetic datasets that are easy to use for end users. All of the dataset synthesis methods included in our benchmark released their generated datasets. However, it may be difficult for users to regenerate these datasets themselves with different augmentations. **For example, [DC, DSA] take days to run for medium to large datasets, [TM] takes a lot of memory and time and requires very specific hyperparameters tuning. KIP uses hundreds of accelerators and requires thousands of GPU hours without open-sourcing their code**. Therefore, instead of encouraging the community to present many options to users, we encourage the community to release **generic high quality datasets** that are easy to use.

---

> ### Author Response · Authors · 2022-08-19
> **Response to reviewer SxsG P3**
>
> **Q6: The transferability and architecture experiments are also a bit unclear to me with respect to the supposed take-away and whether they are “correct” (in the sense of following the assumptions correctly)
> In principle I understood its motivation, but was then puzzled when the text implied (please correct me if I am wrong) that the baseline architecture with which the condensed examples are generated is always the same. Why is this a reasonable assumption for some of the condensation methods, e.g. the ones that are based on very specific trajectories of a particular model, their specific parameters and statistics? The discussion I was lacking in the paper is that even the original papers, like the DC with gradient matching paper, show that in order for condensed examples to be generally useful, one either needs to use a large ensemble of methods to generate them under diverse sets of (potentially random) weights, or suffer from major losses when assuming that the precisely condensed solution is generic. To the best of my knowledge, the trajectory matching paper follows a similar assumption, and uses a large ensemble set of models as the basis. It is unclear to me which of these assumptions were dropped and why this would be reasonable in experimentation.**
>
> R6: Thanks for bringing up this point. The fact that current condensation methods are model-dependent is exactly the motivation of our experiment, i.e. evaluate the condensed dataset on a diverse set of architectures. Our goal is to evaluate condensation methods through the lens of their condensed dataset. Therefore, we follow the settings of the baseline methods for obtaining the condensed dataset and try to encourage the community to develop more architecture-independent synthesis methods.
>
>  In DC with gradient matching, the author claims that “ the condensed images, especially the ones that are trained with convolutional networks, perform well and are thus architecture generic”. Also, they use 1 image per class to evaluate the performance on a very simple dataset(MNIST). In the following work DSA, the same author changes the claim to “synthetic images learned by the convolutional architectures perform best and generalizes to the other convolutional ones”. Again they use the smallest IPC(1) on MNIST. Trajectory matching claims “ Despite being trained for a specific architecture, our synthetic images do not seem to suffer from much over-fitting to that model.”
>
> Therefore, we think all these evaluations are done under limited settings and it cannot really reveal their true performances. So we propose to use our collection of representative models and dataset for better evaluation. Also as suggested by other reviewers as well, we tested the methods with ViT(vision transformer) that’s completely different from CNN. We hope this will provide more insights of synthetic datasets' cross-architecture performance.

---

> ### Author Response · Authors · 2022-08-19
> **Response to reviewer SxsG P2**
>
> **Q3: The data augmentation part is interesting in principle, but it is unclear to me what precise insights are supposed to be drawn from the benchmark as a practitioner.**
>
> R3: Thank you for pointing it out. In order to ensure that our paper conveys the right idea, we have added a “Key Takeaways” section at the end of each benchmark metric (e.g. Augmentation applied at the evaluation phase is critical to the performance of the synthesis dataset). Hopefully that will make it more clear.
>
> To your question that some of the augmentation techniques are overlapping. We allow overlapping augmentations between training and evaluation. The motivation is that dataset users are free to apply any augmentations of their choice based on their requirements. For our benchmark, we only focus on augmentations used in the evaluation phase.  Also although there are some overlapping primitive operations (e.g. Cropping, flipping, etc) between augmentations in our benchmark, the way they are applied is different (e.g. Is it combined with another operation or not or the probability it is applied) and the parameters they use(e.g. The degree of rotation, the size of cropping, etc). Therefore, they may look similar due to shared primitive operations, but they are very different.
>
> **Q4: Whereas some experiments seem to have motivation, others feel more ad-hoc. For instance, what was the hypothesis and expectation behind section 3.3? It is well-known and trivial (from core sets and intuition) that larger amounts of images per class will boost accuracy (figure 2). In core sets, what is then typically investigated is the quality of the selection mechanism, in particular for small sample scenarios. Here however, the dataset condensation methods are all biased by an already picked set of randomly subsampled data instances.**
>
> R4: The purpose of experiments on IPC is not to show that large IPCs will boost accuracy. Rather, the experiment reveals that under large IPCs, the performance of dataset condensation actually degenerates to random selection. We find out that when the IPC goes over 200, none of the synthesis methods are able to learn more information than random selection. We hope this can reveal some useful insights of current synthesis methods and stimulate future research directions.
>
> Regarding the bias introduced by the preselected images. Our experiments show that preselected images won’t introduce a bias that’s large enough to change the relative ranking. If you look at the original papers of DSA, DM and TM . Their reported variances are all less than 1% which aligns with our experiment.
>
>
> **Q5: It would be great to see standard deviations/any measures of statistical deviations across randomly seeded experimental repetitions in any of the experimental result figures. Given that all of the techniques are extremely stochastic, I suspect that the addition of standard deviations would show that there is virtually fairly little difference between the methods in many cases, in particular figure 1 + 2.**
>
> R5: Thank you for your suggestion. We have plotted error bars for all the figures. Although these methods are stochastic, they converge very well(e.g. DC, DSA, DM). We observed close results in multiple runs. The only difference is TM who reports the best results instead of the results after convergence. Regarding the best performance for TM, we are also able to achieve close performances constantly.

---

> ### Author Response · Authors · 2022-08-19
> **Response to reviewer SxsG P1**
>
> **Q1: The choice of methods and taxonomy seems overly narrow. Already when reading lines 28-30 it does not become clear why this particular taxonomy is meaningful. Most importantly, it is unclear to me why “dataset condensation” is framed as a novel and “emerging promising direction” (lines 26). This may indeed be true for the particular idea in the four synthesis method, but only specifically with respect to combining multiple data points/classes into a single instance. If we think of data subsets, core sets have enjoyed rich theoretical investigation and a long history of empirical methods for their selection. It is somewhat surprising to not even have core sets be mentioned, and two oddly specific methods (herding and k-center) be mentioned as some “recent” advocates of data subset selection.**
>
> R1: Thank you for your insightful feedback. We really appreciate it. Our primary goal is to benchmark the newly emerging dataset synthesis methods. Random and k-center are included to establish baselines for DC, rather than being used as representative methods for coreset selection. Based on the benchmark data, we hope to contribute a standardized toolkit and some insights on the synthesis methods and help guide the direction of the field.  As you pointed out, this certainly has caused confusion. Therefore, we have removed the taxonomy and just focus on synthesis methods. But our toolkit can be used by core set methods as well, , and we have released an interface for researchers to add their own methods to the benchmark so potentially any core-set selection method can be easily added. Since it is not very clear to us at this point which core-set method is state-of-the-art, we plan to spend some more time on this and include more core-set selection methods in the future. If the reviewer has any suggestion on any state-of-the-art core-set selection methods please feel free to let us know. Thank you again for your great suggestion.
>
> **Q2: My biggest concern, stemming directly from above argument on the lack of mention to various prior works, is that the framing makes it seem like much of the work is particularly novel and insightful, even if it is at least in parts well known. More precisely, the notion that “kmeans-embed” is some novel proposed method to select data subsets is not only very stretched, but just plain wrong.**
>
> R2: Thank you very much for your suggestion. As you pointed out, core set methods have a long history with much better theoretical guarantees. Our goal is not to compare core set methods with synthesis methods, but to benchmark and guide synthesis methods development and only include commonly used basic selection based methods as naive baselines. The reason we mention Kmeans-emb is that in all previous works(DC,DSA,DM,TM ), K-center performs even worse than random selection. We did a few rounds of analysis and found out the root cause: the model they use to generate the feature embeddings are trained for too many epochs VS just 1 epoch used by us. And it achieves better performance than most other selection based methods compared in previous condensation works. This is why we included it from the very beginning.
>
> But as you pointed out, this is not novel work. We just reestablished a correct baseline for K-Center which has existed for quite a long time. Therefore we have taken your advice and removed core set methods from the definition of condensation methods and renamed our implementation of Kmeans-emb just to K-Center. We want to thank you again for your great suggestion.

---

### Official Review · Reviewer_aF85 · 2022-07-27
**An Organized and Comprehensive Approach to Evaluating Dataset Condensation**

**Rating:** 7
**Confidence:** 3
**Clarity:** The paper is written clearly and easy…

**Strengths:**

* A thoughtful and comprehensive benchmark for evaluating dataset condensation methods
* Strong analysis and experimental studies on existing condensation methods
* Convenient and well-documented API to easily access the benchmark

**Weaknesses:**

* The datasets (CIFAR10/100 and TinyImageNet) and architectures (MLP/ConvNets/ResNets) seem a little bit limited in scope for understanding transferability across architectures. If the goal is to test condensed datasets in *different* settings, it makes more sense to consider architectures far beyond CNNs

**Additional Feedback:**

* Why choose NAS as the sole downstream task within the benchmark? E.g. why not also consider the effect of dataset condensation on continual learning?

**Correctness:**

The benchmark is defined appropriately and analyses are carried out without error.

**Documentation:**

Yes, https://github.com/justincui03/dc_benchmark provides the necessary scripts and datasets to reproduce the results in the paper.

**Ethics:**

There are no ethical concerns.

**Relation To Prior Work:**

The authors note that prior work on dataset condensation only consider limited settings when evaluating their methods, and that this is the first comprehensive, multi-dimensional benchmark to study dataset condensation.

**Summary And Contributions:**

This work aims to provide a comprehensive benchmark and library to study the true effectiveness of dataset condensation methods. In particular, it motivates and provides a framework for carefully studying:
* Model performance under various data augmentations on the condensed dataset
* Model performance under a wide range of dataset compression ratios
* Dataset condensation effects across model architectures (MLPs, ConvNets, and ResNets)
* Speedup effects on NAS

Thorough evaluation on the benchmark using common architectures and dataset condensation methods is provided.

---

> ### Author Response · Authors · 2022-08-19
> **Response to reviewer aF85**
>
> **Q1. The datasets (CIFAR10/100 and TinyImageNet) and architectures (MLP/ConvNets/ResNets) seem a little bit limited in scope for understanding transferability across architectures. If the goal is to test condensed datasets in different settings, it makes more sense to consider architectures far beyond CNNs**
>
>
> R1: Thanks for the great suggestion. We included ViT as part of our benchmark that is very different from CNNs and reported the performance in our work.
>
> **Q2. Why choose NAS as the sole downstream task within the benchmark? E.g. Why not also consider the effect of dataset condensation on continual learning?**
>
> R2: Since dataset condensation aims at accelerating model training, it is essential to study how the condensed dataset effects model training. From this perspective, we deem NAS as the principled systematic method for evaluation.
>
> Also NAS evaluation was performed in some of the works. However, different data condensation papers apply different search space and search methods for NAS which makes it hard to compare the results between different papers. In the NAS community there are already standardized benchmarks such as NAS-Bench-201. So we think it is clear that the condensation community should also compare their methods on this benchmark to have a fair comparison between different methods. We want to encourage the community to use standardized reproducible benchmarks to evaluate it.
>
> We agree that continual learning is another application of data condensation. However, since the algorithms for continual learning are still evolving quickly in recent years and we need more time to investigate what's the standard benchmark and method used in continual learning. Therefore, we plan to add continual learning as well as other tasks in the future. Thank you again for your suggestion!

---

### Official Review · Reviewer_Qbsw · 2022-07-27
**A good benchmark to establish and standardize a new field of research**

**Rating:** 7
**Confidence:** 3
**Correctness:** The claims of the submission are corr…
**Clarity:** The paper is very well written and cl…

**Strengths:**

- The benchmark is modular and standardizes the evaluation of dataset condensation methods
- The benchmark covers the primary use cases of condensed benchmarks
- The experimental section evaluates the state of the art condensation methods and identifies important trends across all methods.
- The code is open source and seems easy to use
- There is an established scoreboard for the benchmark which will foster continued improvement


**Weaknesses:**

- The benchmark only focuses on 3 image classification datasets, therefore the benchmark won’t indicate the usefulness of these techniques on other tasks or domains
- The benchmark should include some indication/evaluation of any bias introduced by the dataset condensation method.


**Additional Feedback:**

I think this can be a very useful benchmark if the authors involve the community and enable others to submit their solutions without substantial effort.

**Documentation:**

The documentation on the Github page could be improved to give specific instructions on how to implement a new solution and submit a score.

**Ethics:**

As mentioned in the weaknesses section, I believe it should be discussed how dataset condensation can introduce biases into a dataset that may go undetected in normal model evaluation.

**Relation To Prior Work:**

The paper does not have a prior work section but it is the first benchmark of this field to my knowledge and it evaluates the current state of the art in dataset condensation.

**Summary And Contributions:**

The paper presents a benchmark for fairly comparing dataset condensation methods. The benchmark evaluates a condensation method’s performance with multiple data augmentation schemes and compression ratios, in addition to its performance in neural architecture search and cross-architecture settings.

---

> ### Author Response · Authors · 2022-08-19
> **Response to reviewer Qbsw**
>
> **Q1: The benchmark only focuses on 3 image classification datasets, therefore the benchmark won’t indicate the usefulness of these techniques on other tasks or domains**
>
> R1: Thank you for pointing it out. Other reviewers raised the same question. Most of the recent works on dataset condensation focus on image classification. Initially,  we considered other vision tasks such as object detection and image segmentation. However, due to data unavailability issues, we will have to push the evaluation in the future version. The reason for that is to evaluate such tasks, we will need bounding boxes in the condensed dataset, which requires additional human labeling work. In the future, if there are new datasets and tasks for this field, we will update the results in our github page.
>
> We also considered tasks in other domains such as NLP. One related work in this domain is [Soft-Label Dataset Distillation and Text Dataset Distillation](https://arxiv.org/pdf/1910.02551.pdf). Although there are some shared properties, we believe it’s worth its own benchmark. Therefore, in our first version we decided to focus on image classification tasks only and introduce new tasks and domains in our future versions.
>
> **Q2: The benchmark should include some indication/evaluation of any bias introduced by the dataset condensation method.**
>
> R2: This is a great point. As dataset synthesis is a newly emerging technique, as far as we know, there is no systematic bias evaluation protocol defined for the field. This is definitely an area that’s worth spending lots of effort into. At the same time, we have added one simple analysis for the distribution of synthetic images to see if there are any biases in terms of data distribution. Please see the “Distribution bias of synthetic dataset” section in our updated appendix.
>
> **Q3: The paper does not have a prior work section**
>
> R3: Thanks for the suggestions. In our latest version, we made a separate “related work” section including prior works, and a short mathematical summary. Hope it will make the paper easier to read.
>
> **Q4: Add a method to enable others to submit their solutions without substantial effort**
>
> R4: Thank you very much for your suggestion. Involving the community is one of our primary goals. We have updated our leaderboard(https://dc-bench.github.io/) with detailed instructions  for submitting new methods easily. At the same time, we will also keep track of the new works in the domain and keep updating the leaderboard and benchmark proactively.  Even since our github repo is online, we have received multiple requests from the authors to include their work in our benchmark.
>
> **Q5: The documentation on the Github page could be improved to give specific instructions on how to implement a new solution and submit a score.**
>
> R5: Thank you for pointing it out. We have updated our github page to include detailed instructions on how to add a new method/dataset/augmentation/model. Also as mentioned above, we updated instructions on how to submit a synthetic dataset and score. Here are the links to our benchmark(https://github.com/justincui03/dc_benchmark) and leaderboard(https://dc-bench.github.io/).

---

> > ### Comment · Reviewer_Qbsw · 2022-08-26
> > **Response**
> >
> > Thank you for addressing my concerns. I believe the paper is worthy of acceptance in its current form.

---

> > > ### Author Response · Authors · 2022-08-26
> > > **Thank you for your reply**
> > >
> > > We thank the reviewer for going through the responses. We are very glad that your questions and concerns are addressed in our updates. If any further questions come up, please feel free to let us know, and we are more than happy to discuss them with you. Thank you again for your time and effort!

---

### Official Review · Reviewer_p7vP · 2022-07-28
**A benchmark for evaluating dataset shrinkage methods**

**Rating:** 4
**Confidence:** 3
**Clarity:** Paper is poorly structured and hard t…

**Strengths:**

1. The paper attempts to classify the conditions under which the training sample compression method should work well.
2. The methods tested in the paper are state-of-the-art.

**Weaknesses:**

1. Related work is described in introduction and benchmark description. That makes the paper hard to comprehend. I believe that separating related work in its' own section would benefit the paper.
2. The benchmark uses CIFAR 10 and 100, Tiny ImageNet as data sources and is limited to image classification.
3. Metrics used for evaluation are described poorly.
4. In general, the work does not look like a complete benchmark with a clear testing procedure. It is rather a set of experiments with different methods.

**Additional Feedback:**

None.

**Correctness:**

It seems to me that the work should be restructured so that there is a clearer separation between the data, the testing procedure (including metrics), and the methods being tested.

**Documentation:**

The benchmark seems to be easily used via accompanied code.

**Ethics:**

No ethics concerns.

**Relation To Prior Work:**

The differences from previous dataset condensation evaluation procedures are sufficiently described in the introduction.

**Summary And Contributions:**

The paper proposes a dataset and procedure for testing methods that reduce the training sample. The data sources are the well-known CIFAR and TinyImageNet datasets. The authors test how the existing methods behave when using different augmentations, compression ratios, neural network classifiers (including those found with NAS).

---

### Author Response · Authors · 2022-08-25
**Summary of Our Responses**

We thank our reviewers for their valuable feedback. We appreciate that the reviewers commented positively on the relevance(R1, R2, R4, R6), presentation(R3, R4), extensiveness(R3, R5) and contribution to the community(R1, R2, R3, R4, R5, R6).

**We have received several inquiries and interests in this benchmark from the community, including requests from authors to incorporate their newly developed methods, or help to collaborate on further developing the benchmark. We are working very closely with the community on these requests.**

We’d also like to acknowledge the insightful questions and concerns that our reviewers raised and that the reviewers explicitly granted us an opportunity to elaborate on these points.

1. Add a related work section for easier reading(Section 2)
2. Remove the taxonomy for selection based methods and synthesis based methods. **Our paper mainly focuses on data synthesis methods**.
3. Update figure 1,2,3,4 and the figures in appendix with error bar
4. Add key takeaways(Section 4) to make our insights more clear for practitioners.
5. Add metrics summary(Section 3) for better understanding
6. Include a new method KIP[31, 32] to broaden our included methods.
7. Rename Kmeans-emb back to K-Center and remove the novelty claim.
8. Update our codebase and leaderboard with better documentation and easier ways to incorporate new methods/datasets/models, etc.


Since we haven't heard back from our reviewers, we would like to reach out to see if our responses have addressed your questions and concerns? As the discussion period is close to an end, if you have any further questions, we are more than happy to discuss them further.

Please kindly let us know your feedback. Thank you for your time and help!

---

### Meta-Review · Area_Chair_WG8m · 2022-09-16

**Recommendation:** Accept
**Confidence:** 4

**Metareview:**

There are quite a few problems raised by the reviewers worth paying attention to:

- Thoroughness of evaluations: Performance metrics are difficult to understand and insufficiently justified and described. There's also mention of non-performance related metrics like bias not being adequately considered. There's some debate over the appropriateness of the included baselines, and some skepticism about the justification for some experiments. Furthermore, Reviewer SxsG notes, "It would be great to see standard deviations/any measures of statistical deviations across randomly seeded experimental repetitions in any of the experimental result figures" and I agree. I also think authors could do a better job in the main text or supplement justifying the use of these specific baselines. However, in the revised version of the paper, many of these issues are addressed with the re-write of Section 3.

- Restricted scope: Many mention the limitation of focusing on images and a small handful of datasets. Reviewer aF85 notes, "The datasets (CIFAR10/100 and TinyImageNet) and architectures (MLP/ConvNets/ResNets) seem a little bit limited in scope for understanding transferability across architectures" and I'd agree, though I recognize that a goal of the tool is to allow for others to also contribute datasets, and send their models to be tested. Also, the authors are correct in noting that many of these issues are a byproduct of the fact that much of the data condensation work so far has been focused on image datasets.

- Usability: Reviewer Qbsw, Reviewer SxsG and especially Reviewer 5Lwd all mention issues with documentation, and not how difficult it is to follow the provided instructions in order to assess a model, add a new dataset, etc. However, the authors seem to have addressed many of the concerns, improving documentation significantly following this feedback.

Overall, it seems the authors paid attention to reviewer critiques and responded respectfully and meaningfully to the provided feedback. Given the importance of the topic and the current lack of testing infrastructure in this area, I recommend we accept this paper as a poster. I hope authors continue to take in feedback at the conference to continue to make further improvements to their benchmarking platform.

---

### Decision · Program_Chairs · 2022-09-16

Accept